# Cobalt ion interaction with TMEM16A calcium-activated chloride channel: Inhibition and potentiation

**Dung M. Nguyen[1], Louisa S. Chen[1], Grace Jeng[1], Wei-Ping Yu[1], Tsung-Yu Chen[1,2]\***

**1** Center for Neuroscience, University of California, Davis, California, United States of America,
**2** Department of Neurology, University of California, Davis, California, United States of America

\* tycchen@ucdavis.edu

**Data Availability Statement:** All relevant data are within the paper and its Supporting Information files.

## Abstract

TMEM16A, a $Ca^{2+}$-sensitive $Cl^-$ channel, plays key roles in many physiological functions related to $Cl^-$ transport across lipid membranes. Activation of this channel is mediated via binding intracellular $Ca^{2+}$ to the channel with a relatively high apparent affinity, roughly in the sub-μM to low μM concentration range. Recently available high-resolution structures of TMEM16 molecules reveal that the high-affinity $Ca^{2+}$ activation sites are formed by several acidic amino acids, using their negatively charged sidechain carboxylates to coordinate the bound $Ca^{2+}$. In this study, we examine the interaction of TMEM16A with a divalent cation, $Co^{2+}$, which by itself cannot activate current in TMEM16A. This divalent cation, however, has two effects when applied intracellularly. It inhibits the $Ca^{2+}$-induced TMEM16A current by competing with $Ca^{2+}$ for the aforementioned high-affinity activation sites. In addition, $Co^{2+}$ also potentiates the $Ca^{2+}$-induced current with a low affinity. This potentiation effect requires high concentration (mM) of $Co^{2+}$, similar to our previous findings that high concentrations (mM) of intracellular $Ca^{2+}$ ($[Ca^{2+}]_i$) can induce more TMEM16A current after the $Ca^{2+}$-activation sites are saturated by tens of μM $[Ca^{2+}]_i$. The degrees of potentiation by $Co^{2+}$ and $Ca^{2+}$ also roughly correlate with each other. Interestingly, mutating a pore residue of TMEM16A, Y589, alters the degree of potentiation in that the smaller the sidechain of the replaced residue, the larger the potentiation induced by divalent cations. We suggest that the $Co^{2+}$ potentiation and the $Ca^{2+}$ potentiation share a similar mechanism by increasing $Cl^-$ flux through the channel pore, perhaps due to an increase of positive pore potential after the binding of divalent cations to phospholipids in the pore. A smaller sidechain of a pore residue may allow the pore to accommodate more phospholipids, thus enhancing the current potentiation caused by high concentrations of divalent cations.

## Introduction

The TMEM16 gene family consists of two types of transmembrane proteins with distinct molecular functions: ion channels and phospholipid scramblases [1, 2]. TMEM16A, the first identified member in this gene family [3–5], is a $Ca^{2+}$-activated $Cl^-$ channel. It is highly

**Funding:** This work was supported by an NIH grant R01GM065447. The funders had no role in study design, data collection and analysis, decision to publish, or preparation of the manuscript.

**Competing interests:** The authors have declared that no competing interests exist.

expressed in the apical membrane of ductal epithelial cells, and one well-documented physiological function of this channel known for decades is its critical roles in transepithelial $Cl^-$ transport [6]. Another $Ca^{2+}$-activated $Cl^-$ channel from the TMEM16 family is TMEM16B, which helps transduce odor stimuli into electrical signals in olfactory receptor neurons [7, 8]. Physiologically, these two anion channels conduct $Cl^-$ across the lipid membrane in response to sub-µM or low µM concentrations of intracellular $Ca^{2+}$ ([$Ca^{2+}$]$_i$). The other type of TMEM16 family members, such as fungus nhTMEM16, afTMEM16 and mammalian TMEM16F, are phospholipid scramblases. One documented physiological role of TMEM16F is to scramble phospholipids in platelet cell membranes and thus expose phosphatidylserine to the extracellular environment, an important step in the signaling cascade for blood coagulation [9–11]. A defect in the scrambling activity of TMEM16F has been known to result in a bleeding disorder called Scott syndrome [10, 12]. Interestingly, activation of phospholipid scramblases by [$Ca^{2+}$]$_i$ not only leads to scrambling of membrane phospholipids but also causes ionic conduction across lipid membranes. The physiological roles of the current conduction in these phospholipid scramblases remain to be identified.

Although the physiological functions of ion channels and phospholipid scramblases appear to be quite different, the structures of these two types of TMEM16 proteins are similar to each other [13–18]. These TMEM16 molecules are homodimeric proteins [19], with 10 transmembrane helices (helix 1–10) present in one subunit. In each subunit, helices 3–8 form a conduit thought to be the pathway for the substrate (ions or phospholipids) transport across lipid membranes (Fig 1). Structural studies of fungus TMEM16 proteins (such as nhTMEM16 and afTMEM16) show that the conduit thought to be the substrate-transport pathways appears as an open groove [14, 16], raising the possibility that phospholipid transport may resemble swiping a credit card through a card reader [11, 14, 20–22]. However, recent cryo-EM studies on the mammalian TMEM16F protein revealed that the extracellular half of helix 4 (the helix colored in orange in Fig 1) slants more towards the transport pathway than the corresponding helix in fungus scramblases [13, 23], and therefore the extracellular half of the transport pathway appears to be fully enclosed by helices 3–8 (Fig 1A). The intracellular half of the transport pathway in mammalian TMEM16F, however, is still open, making room for membrane phospholipids to contribute to the wall-lining. This structure of the transport pathway in mammalian TMEM16F, namely, an enclosed ion-permeation conduit only in the extracellular half of the pathway is shared by TMEM16A (Fig 1B), which functions as a convential ion channel but not a phospholipid scramblase [15, 17, 18]. The implication from the pore structures of various TMEM16 molecules with respect to their abilities of transporting phospholipids remains to be determined [13, 23].

In addition to the transport pathways for anions and phospholipids, the $Ca^{2+}$-activation sites have also been identified in the high-resolution structures of TMEM16 molecules [13–18, 23]. In both types of TMEM16 molecules, each protein subunit possesses two $Ca^{2+}$-binding sites, in which several acidic residues use their sidechain carboxylates to coordinate the bound $Ca^{2+}$. Functional studies have shown that activation of one subunit of TMEM16A still generates a dose-response activation curve with a Hill coefficient greater than unity [24], consistent with the structural findings that multiple $Ca^{2+}$ can bind to the activation sites of a subunit and that one subunit contains one pore. Mutations of these acidic residues dramatically alter the half-effective $Ca^{2+}$ concentration ([$Ca^{2+}$]) in activating the channel—mutation of a single glutamate or aspartate residue increases the intracellular [$Ca^{2+}$] ([$Ca^{2+}$]$_i$) required for channel activation by three orders of magnitude [24–26]. Thus, coordination of $Ca^{2+}$ in the channel activation sites by multiple carboxylates from these acidic residues underlies the relatively high apparent affinity of $Ca^{2+}$ in activating the channel. In TMEM16A, it has been shown that $Ca^{2+}$ binding to the activation sites can also have an effect on ion permeation—the binding reduces

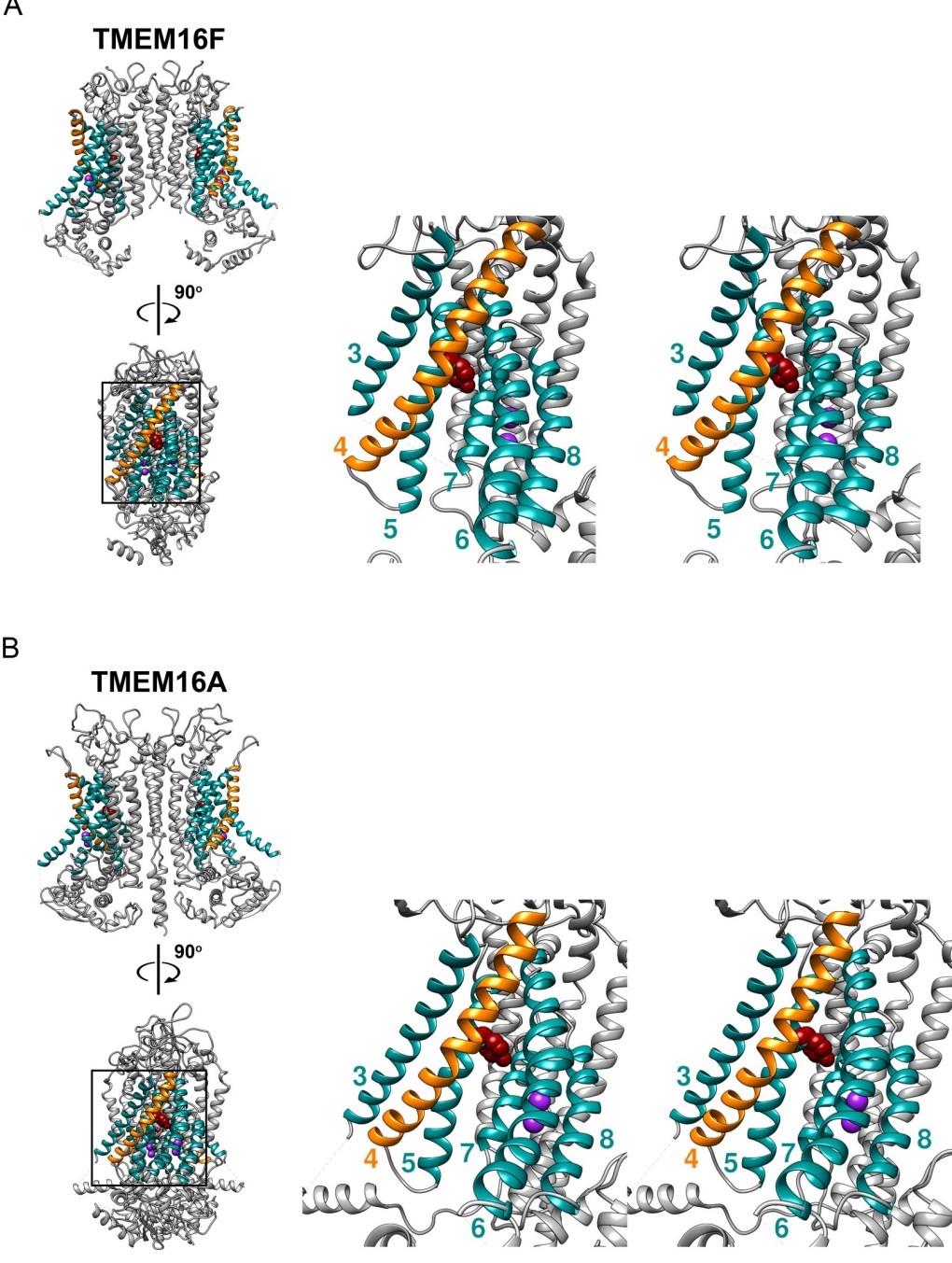

**Fig 1. Structures of TMEM16A and TMEM16F with bound Ca²⁺.** (A) TMEM16F structure (PDB: 6QPC). (B)
TMEM16A structure (PDB: 5OYB). On the left part of both A & B, the top view is from within the phospholipids for
seeing both subunits. The view below is obtained by rotating the structure 90˚ clockwise around the two-fold axis. The
boxed area in the lower view is expanded on the right using a stereo pair, and the subunit in the back is hidden for
clarity. Purple spheres represent calcium ions, and the space-filled residue in red represents Y589 of TMEM16A or Y564
of TMEM16F. Helices in color are labeled with numbers representing the six transmembrane helices (TM3-8)
encompassing the transport pathway.

the negative charge from these acidic Ca²⁺-coordinating residues, thus altering the rectifica-
tion of the Cl⁻ flux through the channel pore [27]. This is consistent with the structural finding

that the $Ca^{2+}$ activation sites are located not far from the pore. Our previous experiments also showed that altering the sidechain charge of a pore residue affects the rectification of the I-V curve, directly supporting an electrostatic control of $Cl^-$ flux in the pore [24, 28].

While the structural features reasonably explain some of the molecular properties of the TMEM16 proteins, the mechanisms of other functional phenomena of TMEM16 molecules remain unclear. For example, the biphasic dose-response activation curve for the activation of TMEM16A by $Ca^{2+}$ is still puzzling [24, 29]. It is well documented that intracellular $Ca^{2+}$ activates TMEM16A with a half-activation concentration in the range of sub-μM to low μM [5, 30–32]. Therefore, the activation of TMEM16A should be saturated by 10–20 μM $[Ca^{2+}]_i$. Yet, if $[Ca^{2+}]_i$ is increased to hundreds of μM or to mM concentrations, TMEM16A current can be further increased—the current induced by 20 mM $[Ca^{2+}]_i$ was ~30–40% larger than that activated by 20 μM $[Ca^{2+}]_i$ [24, 29]. If the $Ca^{2+}$-activation sites have already been saturated by 20 μM $[Ca^{2+}]_i$, what is the mechanism responsible for the current induced by mM $[Ca^{2+}]_i$? Is the current induced by mM $[Ca^{2+}]_i$ mediated by different $Ca^{2+}$-activation sites or is the mechanism of this low affinity $Ca^{2+}$ effect mediated by the high affinity $Ca^{2+}$-binding sites formed by the aforementioned negatively-charged residues? To further examine the properties of TMEM16A, we initially aimed at using cobalt ions ($Co^{2+}$), to interact with the channel activation sites on this $Cl^-$ channel. We found that intracellular $Co^{2+}$ is a competitive inhibitor for the $Ca^{2+}$ activation of TMEM16A. The experiments also lead to a serendipity finding that high concentrations (hundreds of μM or above) of intracellular $Co^{2+}$ ($[Co^{2+}]_i$) potentiate the $Ca^{2+}$-induced TMEM16A current. The results suggest that the TMEM16A current potentiation by mM $[Co^{2+}]_i$ and that by mM $[Ca^{2+}]_i$ described above may come from the same mechanism, thus explaining the biphasic dose-dependent activation curve of TMEM16A. Interestingly, mutating a pore residue of TMEM16A affects the degree of potentiation, suggesting that the relatively low affinity potentiation site(s) for these two cations may reside in or near the anion transport pathway.

## Materials and methods

### Reagents and cDNA clones

The cDNA of the "*a*" alternative splice variant [32] of the TMEM16A (NCBI reference sequence: NM_001242349.1), subcloned in the pEGFP-N3 or pIRES expression vector (Clontech/Takara Bio), was used throughout the study. The cDNA constructs produced channels with (from the pEGFP-N3 construct) or without (from the pIRES construct) a green fluorescent protein (GFP) attached to the C terminus of the channel proteins. The results obtained from the GFP-tagged and un-tagged TMEM16A were not distinguishable. To create mutations, the QuikChange II Site-Directed Mutagenesis Kit (Agilent Technologies) was used according to manufacturer's instruction. Channel expression was achieved by transiently transfecting the channel cDNAs to human embryonic kidney (HEK) 293 cells using the lipofectamine transfection method [24, 30, 33]. Under an inverted microscope (DM IRB; Leica) equipped with a fluorescent light source and a GFP filter (Chroma Technology), HEK293 cells expressing transfected channels were identified by the green fluorescence from the cells. The chemicals used in this study were all reagent grade. Regular salts such as NaCl and $CoCl_2$ were obtained from MilliporeSigma and MP Medicals. HEPES was obtained from Sigma/Aldrich.

### Electrophysiological methods

Twenty-four to forty-eight hours after transfection, patch-clamp recordings were conducted on HEK293 cells with green fluorescence. All experiments were from excised inside-out membrane patches. The pipette (extracellular) solution contained 140 mM NaCl, 10 mM HEPES,

and 0.1 mM EGTA (pH 7.4). This solution was also used as the intracellular solution containing "zero $Ca^{2+}$." For the intracellular solutions containing specified $[Ca^{2+}]_i$, EGTA was not included because of two considerations. First, including a $Ca^{2+}$ buffer (such as EGTA) in solutions containing both $Ca^{2+}$ and $Co^{2+}$ would alter the free concentration of each ion in a complicated manner. Second, most of the solutions used in this study contained $[Ca^{2+}]_i$ in μM to mM range, which is way beyond the buffering range of EGTA. Without the added EGTA, the free $[Ca^{2+}]_i$ was assumed to be equal to the total added $[Ca^{2+}]_i$. Nonetheless, it should be noticed that when the indicated $[Ca^{2+}]_i$ was low (such as 2 or 5 μM), the real $[Ca^{2+}]_i$ was likely slightly higher because of the contaminating $Ca^{2+}$. Based on the current of a TMEM16F mutant induced by an intraceullar solution containing neither EGTA nor $Ca^{2+}$, the contaminating $[Ca^{2+}]_i$ in this nominal zero-$Ca^{2+}$ solution was less than 1 μM (see S1 Fig). Such a small contaminating $[Ca^{2+}]_i$ should not alter the major conclusions in this paper as most of our experiments were performed on the TMEM16A current induced by much higher $[Ca^{2+}]_i$. All intracellular solutions also contained 10 mM HEPES (pH 7.4) and had a final $[Cl^-]$ of 140 mM. In the experiments of assessing $Co^{2+}$ effects, the intracellular solutions contained $Co^{2+}$ up to 20 mM. Because the source of $Co^{2+}$ was from $CoCl_2$, the concentration of NaCl in the intracellular solution was reduced according to the extra $[Cl^-]$ from the added $[CoCl_2]$. For example, the solution containing 20 mM $[Co^{2+}]$ included 100 mM $[NaCl]$ and 20 mM $[CoCl_2]$. In experiments using high $[Ca^{2+}]_i$, the intracellular $[NaCl]$ was also adjusted accordingly if the total $[CaCl_2]$ in the solution was more than 0.5 mM. These adjustments, likely would alter the ionic strength of the solution, and therefore, the measurements obtained with high divalent cation concentrations (such as 5–20 mM) may be less accurate.

The recording electrodes were made from borosilicate glass capillaries (World Precision Instruments) using a PP830 electrode puller (Narishige). The electrode tip was ~1–2 μm in diameter, and the electrode resistance was between ~1.5 MΩ and ~3 MΩ when filled with the extracellular solution. Voltage clamp experiments were conducted using the Axopatch 200B amplifier (Molecular Devices), and the current was digitized via a Digidata1440 analog-digital signal-converting board controlled by the pClamp10 software (Molecular Devices). Exchanging solutions on the intracellular side of the excised inside-out patch was achieved using the SF-77 solution exchanger (Warner Instruments). Except where indicated, the recording was initiated by stepping the membrane voltage to ±20 mV in the EGTA-containing zero-$Ca^{2+}$ solution. $[Ca^{2+}]_i$ was then applied to open the TMEM16A channel, followed by the application of a particular $[Co^{2+}]_i$ (with the same $[Ca^{2+}]_i$) for 6 sec, which allowed the $Co^{2+}$ effect to reach a steady state. $Co^{2+}$ was then removed by changing the solution back to the one before applying $[Co^{2+}]_i$. Finally, the intracellular $Ca^{2+}$ was removed using the EGTA-containing zero-$Ca^{2+}$ solution. In evaluating the kinetics of the current activation or deactivation upon applying or removing $[Ca^{2+}]_i$ (such as those in Fig 4), the recordings were made in the presence of the indicated $[Co^{2+}]_i$ throughout the whole recording course. For these experiments, $[Co^{2+}]_i$ was first applied, and the voltage was then stepped to ±20 mV followed by applying and then removing $[Ca^{2+}]_i$ in the presence of the same $[Co^{2+}]_i$.

To assess the current potentiation by high $[Ca^{2+}]_i$, we employed a three-pulse protocol described previously [30]. The cytoplasmic side of the patch was sequentially exposed to a control solution containing 20 μM $[Ca^{2+}]_i$, a test solution containing $[Ca^{2+}]_i$ from 50 μM to 20 mM, and then again the 20 μM $[Ca^{2+}]_i$ solution. We used this experimental protocol to minimize the effect of current rundown so that a dose-response effect of the test $[Ca^{2+}]_i$ can be more precisely evaluated. The current induced by the test $[Ca^{2+}]$ was then normalized to the average of the two currents obtained in 20 μM $[Ca^{2+}]$, and the normalized values were used for generating dose-response curves.

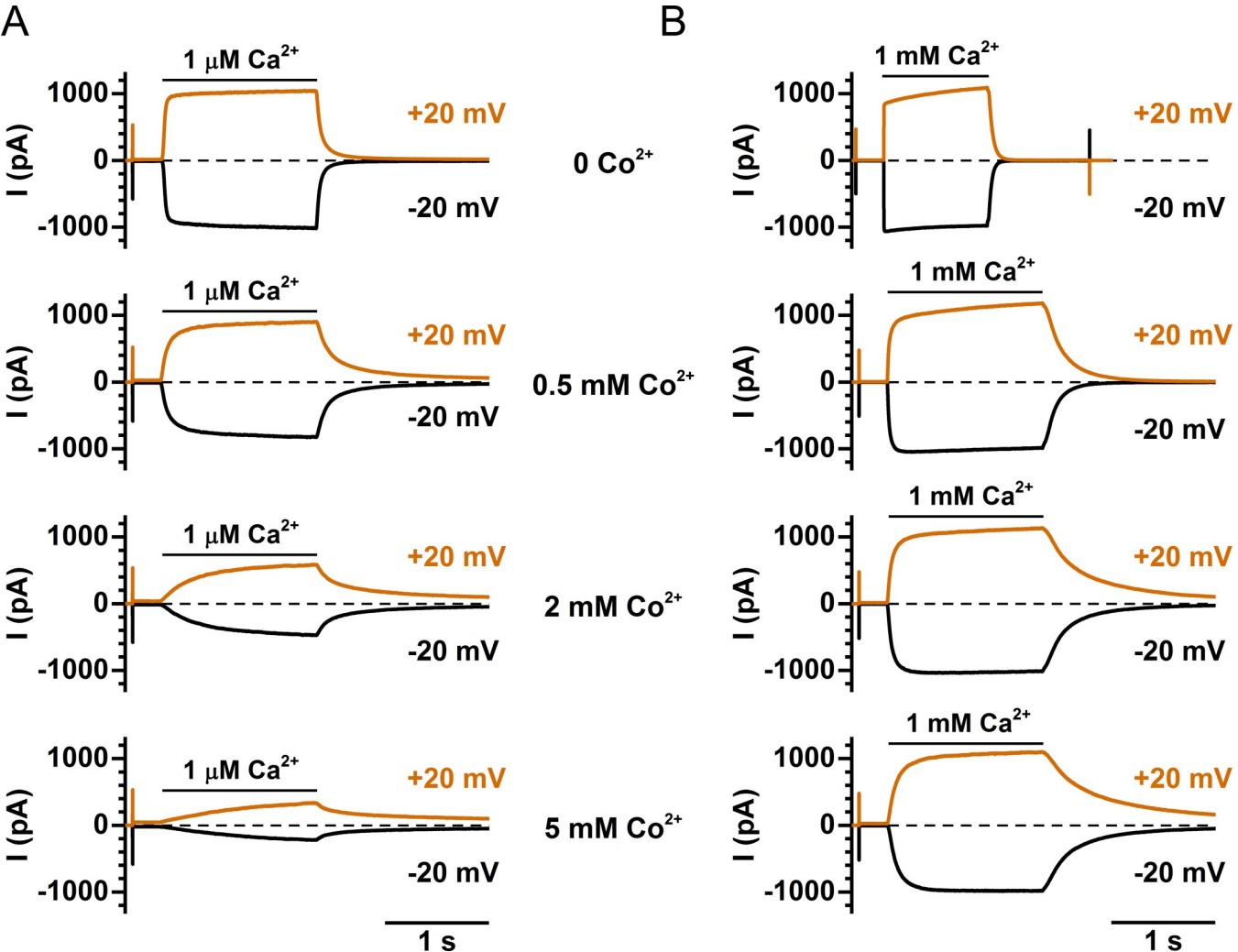

**Fig 4. Effects of Co$^{2+}$ on the kinetics of TMEM16A activation/deactivation.** The indicated $[Co^{2+}]_i$ was present throughout the entire recording period in all recordings. TMEM16A current was induced by (A) 1 μM $[Ca^{2+}]_i$ and (B) 1 mM $[Ca^{2+}]_i$. All recording traces in A were from the same patch, while the recording traces in B were from another patch. Notice intracellular Co$^{2+}$ slows down the current activation upon applying $[Ca^{2+}]_i$ and the current reduction upon $[Ca^{2+}]_i$ washout in a dose-dependent way. At the same time Ca$^{2+}$ antagonizes the effects of Co$^{2+}$ on the activation/deactivation kinetics. Similar results were observed in another 4 patches (two patches each with 1 μM $[Ca^{2+}]_i$ and 1 mM $[Ca^{2+}]_i$).

## Data analysis

We analyzed the experimental data using the combination of pClamp (Molecular Devices) and Origin (OriginLab, Co.) software. For data analyses, the background leak current in the absence of $[Ca^{2+}]$ was first subtracted from the current obtained in the presence of $[Ca^{2+}]_i$. Upon the application of $[Co^{2+}]_i$, the Ca$^{2+}$-induced current of TMEM16A was potentiated almost immediately followed by a slower process of inhibition. We named $I_0$ as the control current immediately before the application of $[Co^{2+}]_i$. The peak of the current potentiation was defined as $I_{peak}$, while the current measured at the end of the 6-sec Co$^{2+}$ application was defined as $I_{Co}$. We performed the same experiments on the wild-type (WT) TMEM16A channel as well as on more than 10 point mutants of residue Y589. To evaluate the degrees of potentiation and inhibition, a Co$^{2+}$ potentiation coefficient was calculated by dividing $I_{peak}$ by $I_0$, while a Co$^{2+}$ inhibition coefficient was calculated by $I_{Co}/I_{peak}$. All averaged results are presented as mean ± S.E.M.

To analyze the dependence of the degree of inhibition on $[Co^{2+}]$, the values of $I_{Co}/I_{peak}$ were plotted against $[Co^{2+}]_i$, and the dose-dependent $Co^{2+}$ inhibitions were fitted to a Langmuir equation to evaluate the apparent $Co^{2+}$ affinity:

$$I_{norm} = \frac{I_{Co}}{I_{peak}} = \frac{1}{1 + \frac{[Co^{2+}]}{K_{1/2}}} \tag{1}$$

where $K_{1/2}$ is the fitted half-inhibition concentration. For the current potentiation, because the potentiation effect appeared to be voltage dependent and because the effect of the highest $[Co^{2+}]_i$ (20 mM) at +20 mV appeared unsaturated, we did not fit the $[Co^{2+}]_i$-dependent potentiation to any type of binding isotherm equations. The data points were connected by line segments.

To evaluate the apparent $Ca^{2+}$-dissociation rate from the $Ca^{2+}$-activation site, the time course of the current reduction upon the final washout of $[Ca^{2+}]_i$ was fit to a single-exponential decay function:

$$I(t) = (I_{peak} - I_{Co}) \times \left(1 - \exp\left(-\frac{t}{t_{off}}\right)\right) \tag{2}$$

where $I(t)$ is the TMEM16A current at time t and the meaning of $I_{peak}$ and $I_{Co}$ are as defined above. The time constant of the current reduction process ($\tau_{off}$) was used to correlate with the degrees of $Co^{2+}$ inhibition or $Co^{2+}$ potentiation. Two other parameters, the sidechain hydrophobicity and the molecular volume of the amino acid placed at position 589, were also used to correlate with the $Co^{2+}$ inhibition or the $Co^{2+}$ potentiation. The values of the sidechain hydrophobicity and the molecular volume of the amino acid were obtained, respectively, from Kyte and Doolittle [34] and Zamyatnin [35].

## Results

Examples of the intracellular $Co^{2+}$ effects on the WT TMEM16A are shown in Fig 2. Intracellular $Co^{2+}$ (up to 20 mM) by itself does not activate any current in WT TMEM16A (Fig 2A, left panel), while a robust current can be induced by intracellular $Ca^{2+}$ from the same membrane patch (Fig 2A, right panel). However, as illustrated by the recording traces shown in Fig 2B, intracellular $Co^{2+}$ (from 50 μM to 5 mM shown in these traces) appears to have dual effects on the $Ca^{2+}$-induced TMEM16A current. In these recordings, TMEM16A currents were first activated by 30 μM $[Ca^{2+}]_i$. Upon applying $[Co^{2+}]_i$ (in the presence of 30 μM $[Ca^{2+}]_i$), the currents were first potentiated, then inhibited, although the potentiation of the current by 50 μM $Co^{2+}$ was not observed. It appears that the degrees of the current potentiation and inhibition depend on $[Co^{2+}]_i$.

To understand the $Co^{2+}$ effects on TMEM16A, we first examined the mechanism of inhibition. Fig 3A shows $Co^{2+}$ inhibition on the TMEM16A current activated by different saturating concentrations of $[Ca^{2+}]_i$ (from 10–300 μM). Although $[Co^{2+}]_i$ in all these recordings is the same (2 mM), the degree of inhibition decreases with an increase of $[Ca^{2+}]_i$ used for activating the current. Fig 3B shows that at both +20 mV (upper panel) and -20 mV (lower panel), the $K_{1/2}$ of $[Co^{2+}]_i$-dependent inhibition (shown in Table 1) was shifted in parallel to higher values by increased $[Ca^{2+}]_i$. These results suggest that $Co^{2+}$ may inhibit the current by competing with $Ca^{2+}$ for the activation sites. This competition can also be directly appreciated from the kinetics of TMEM16A current induction by $[Ca^{2+}]_i$. Fig 4A and 4B show two sets of the TMEM16A current activation time course in the presence of 0, 0.5, 2, and 5 mM $[Co^{2+}]_i$. Both the rate of the current activation upon applying $[Ca^{2+}]_i$ and that of the current reduction after

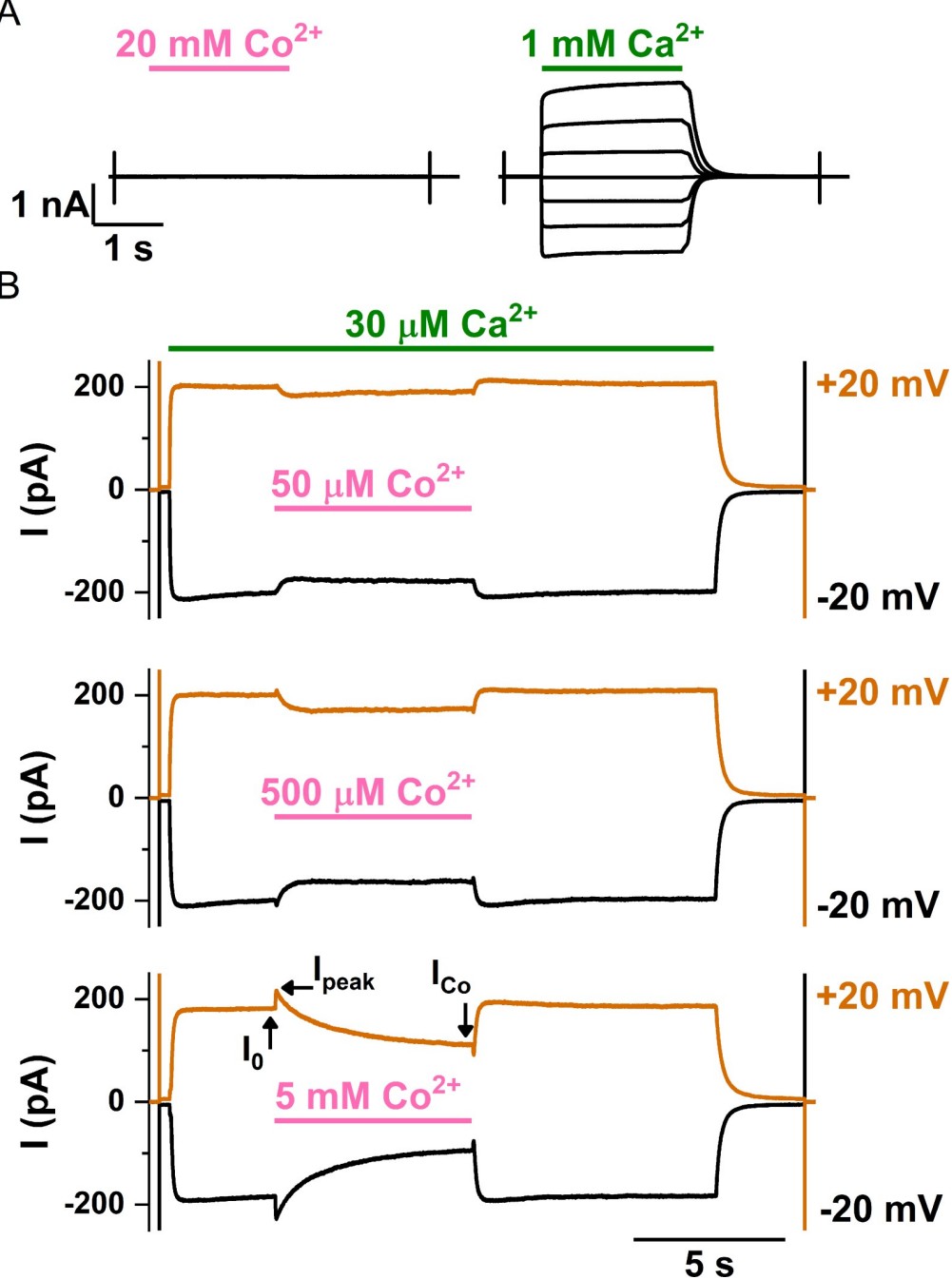

**Fig 2. Effects of Co²⁺ on the WT TMEM16A channel.** (A) Effects of intracellular Co$^{2+}$ (left) and Ca$^{2+}$ (right) when applied alone. The membrane voltage began at 0 mV, and was then stepped from -60 mV to +60 mV in +20 mV steps. $[Co^{2+}]_i$ (pink line) and $[Ca^{2+}]_i$ (green line) were applied as indicated on the same membrane patch. (B) Effects of various $[Co^{2+}]_i$ on the TMEM16A current activated by 30 μM $[Ca^{2+}]_i$. The membrane voltage was 0 mV at the start of the recording and then stepped to -20 mV (black traces) and +20 mV (orange traces), respectively, at the time indicated by the capacitive spikes. $[Ca^{2+}]_i$ (30 μM) was then applied, followed by applying Co$^{2+}$ in the presence of the same $[Ca^{2+}]_i$ as indicated by horizontal lines. $I_{peak}$, $I_0$ and $I_{Co}$ are defined as in the Data Analysis section in METHODS.

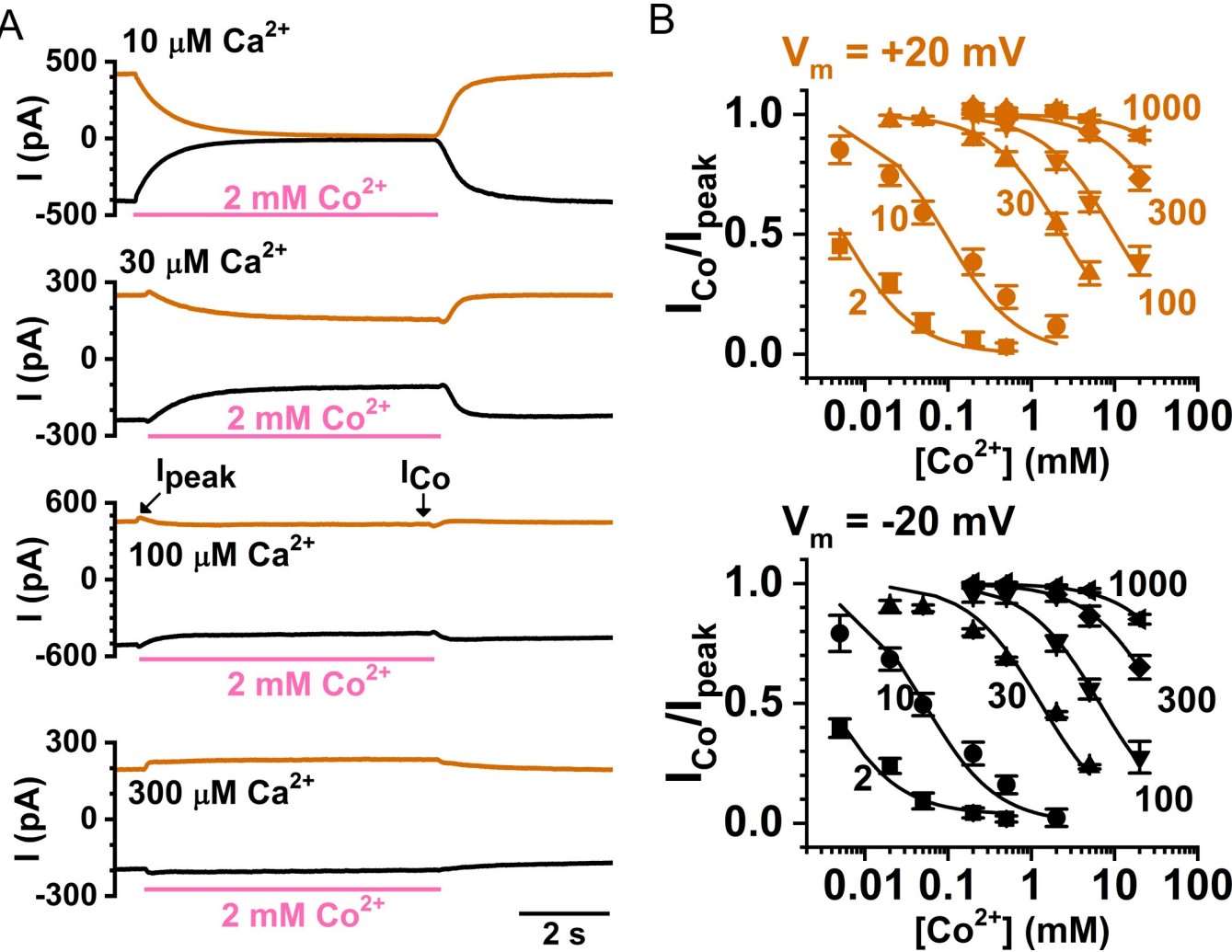

**Fig 3. Competitive inhibition by Co²⁺ on [Ca²⁺]ᵢ-induced TMEM16A current.** **(A)** Inhibition by 2 mM Co²⁺ of the TMEM16A current activated by various [Ca²⁺]ᵢ at -20 mV (black) and +20 mV (orange). **(B)** Concentration-dependent Co²⁺ inhibition of the TMEM16A current activated by various [Ca²⁺]ᵢ at +20 mV (upper panel) and -20 mV (lower panel). Numbers next to each individual dose-response curves represent the [Ca²⁺]ᵢ (in μM) used in activating the current. Data points were fitted to the Langmuir function defined in METHODS (Eq 1). The fitted $K_{1/2}$ values of Co²⁺ inhibition at -20 mV and +20 mV in various [Ca²⁺]ᵢ are presented in Table 1.

removing [Ca²⁺]ᵢ are decreased in the presence of [Co²⁺]ᵢ. Thus, Co²⁺ slows down the kinetics of Ca²⁺ binding to the activation sites.

**Table 1. $K_{1/2}$ of Co²⁺ inhibition of the TMEM16A current induced by various [Ca²⁺]ᵢ.**

| [Ca²⁺]ᵢ (μM) | $K_{1/2}$ (mM) of Co²⁺ inhibition | |
|---|---|---|
| | **-20 mV** | **+20 mV** |
| 2 | 0.003 ± 0.001 | 0.005 ± 0.001 |
| 10 | 0.05 ± 0.01 | 0.09 ± 0.02 |
| 30 | 1.3 ± 0.2 | 2.4 ± 0.1 |
| 100 | 6.6 ± 0.4 | 10 ± 1.1 |
| 300 | 36 ± 1.3 | 59 ± 9.6 |
| 1000 | 118 ± 7 | 223 ± 37 |

If $Co^{2+}$ and $Ca^{2+}$ compete for the $Ca^{2+}$-activation sites, the potency of $Co^{2+}$ inhibition may decrease in mutant channels with slower $Ca^{2+}$ dissociation rates. This indeed appears to be the case. Fig 5A depicts recording traces of applying 50 μM, 500 μM, and 5 mM $[Co^{2+}]_i$ to a TMEM16A mutant, Y589A, while Fig 5B shows a comparison of the $[Co^{2+}]_i$-dependent inhibition curves between WT TMEM16A and the Y589A mutant at -20 mV (upper panel) and +20 mV (lower panel), respectively. The Y589A mutant has been reported to have a greater apparent affinity for $Ca^{2+}$ activation than WT channels [15]. Our recordings show that the rate of the current reduction upon $[Ca^{2+}]_i$ removal in this mutant is significantly slower ($\tau_{off}$ of the current reduction process is larger) than in the WT channel (Fig 5C), consistent with a greater $Ca^{2+}$ affinity in this mutant. In comparison with the effect on the WT channel, a higher $[Co^{2+}]_i$ is required to inhibit the Y589A current induced by the same concentration of $[Ca^{2+}]_i$. Meanwhile, the degree of current potentiation by $[Co^{2+}]_i$ is larger in Y589A than in the WT

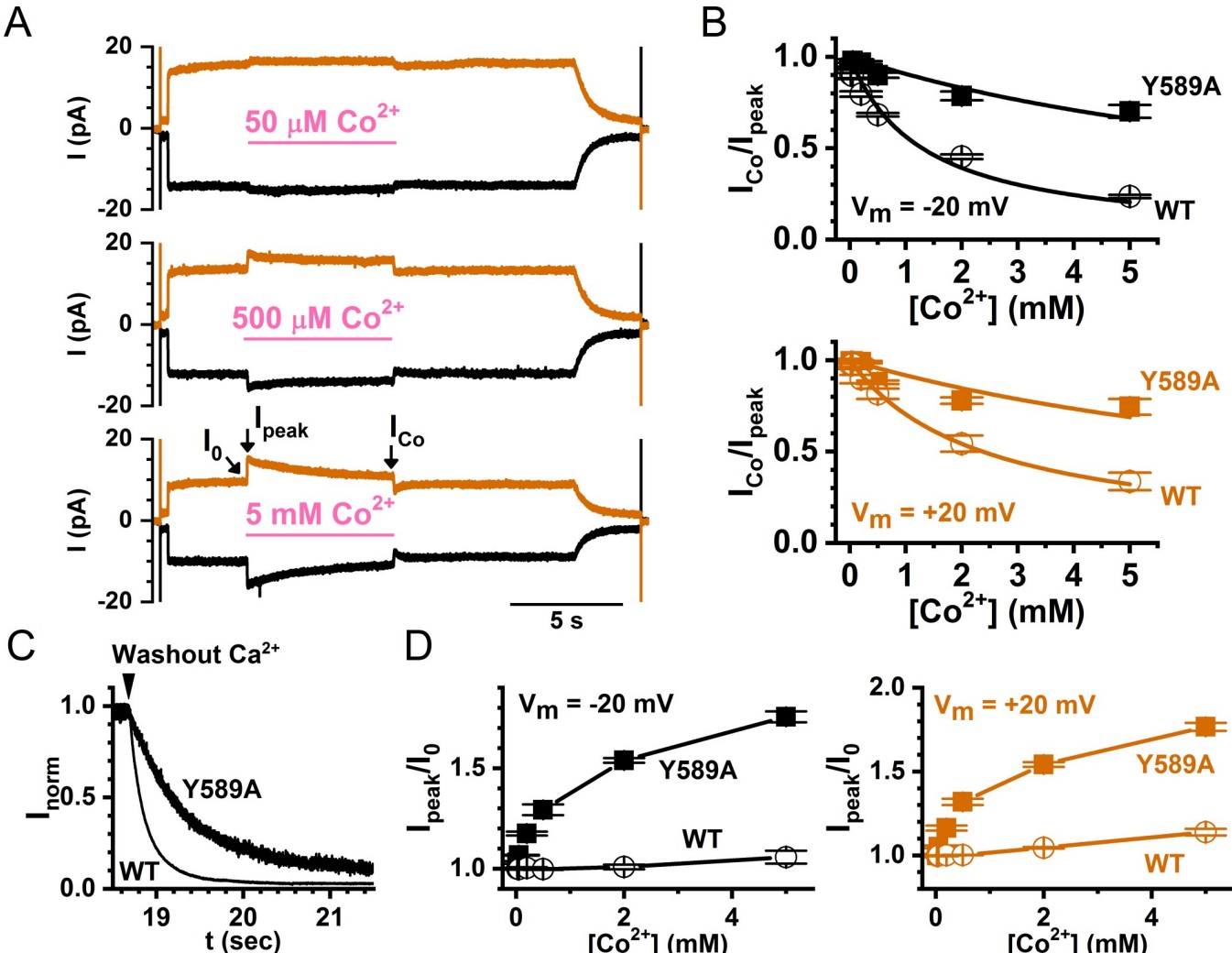

**Fig 5. Comparing $Co^{2+}$ effects between WT TMEM16A and the Y589A mutant.** (A) Effects of various $[Co^{2+}]_i$ on the Y589A mutant. $[Ca^{2+}]_i$ = 30 μM for all patches. (B) Comparison of the dose-response curves of $Co^{2+}$ inhibition between WT TMEM16A and the Y589A mutant. (C) Representative traces comparing the current-reduction process upon washout of $[Ca^{2+}]_i$ at +20 mV between WT TMEM16A and the Y589A mutant. Recorded currents were normalized to the current right before the washout of $[Ca^{2+}]_i$. The current reduction processes were fitted to Eq 2 with the averaged $\tau_{off}$ of 144 ± 5 ms (n = 6) and 601 ± 29 ms (n = 4) for the WT channel and the Y589A mutant, respectively. (D) Comparison of the dose-response curves of $Co^{2+}$ potentiation between WT TMEM16A and the Y589A mutant at -20 mV (left) and +20 mV (right).

channel (Fig 5D). We thus constructed more mutants of Y589 and analyzed $Co^{2+}$ inhibition more extensively. Fig 6 depicts recording traces of the $Co^{2+}$ effects in ten Y589 mutants. These Y589 mutants have different $Ca^{2+}$ dissociation rates as judged from the current reduction time ($\tau_{off}$) after removing $[Ca^{2+}]_i$ near the end of the recordings. For example, the current reduction upon washout of $[Ca^{2+}]_i$ appears slower in Y589G, Y589S, Y589V, and Y589C than in Y589W, Y589H, Y589F and Y589K (Fig 6).

The correlations between $Co^{2+}$ inhibition and the current reduction time ($\tau_{off}$) are shown in Fig 7A and 7B (left panel). Plotting the fraction of remaining current after 2 mM $Co^{2+}$ inhibition ($I_{Co}/I_{peak}$) against the value of $\tau_{off}$ confirms that the potency of $Co^{2+}$ inhibition decreases with the increase of $\tau_{off}$; namely, the slower the $Ca^{2+}$ dissociation rate, the weaker the $Co^{2+}$ inhibition. We also correlated the $Co^{2+}$ inhibition ($I_{Co}/I_{peak}$) obtained at +20 mV (Fig 7A) and at -20 mV (Fig 7B) against two other parameters: the sidechain hydrophobic index (Fig 7A and 7B, middle panel) and the molecular volume (Fig 7A and 7B, right panel) of the amino

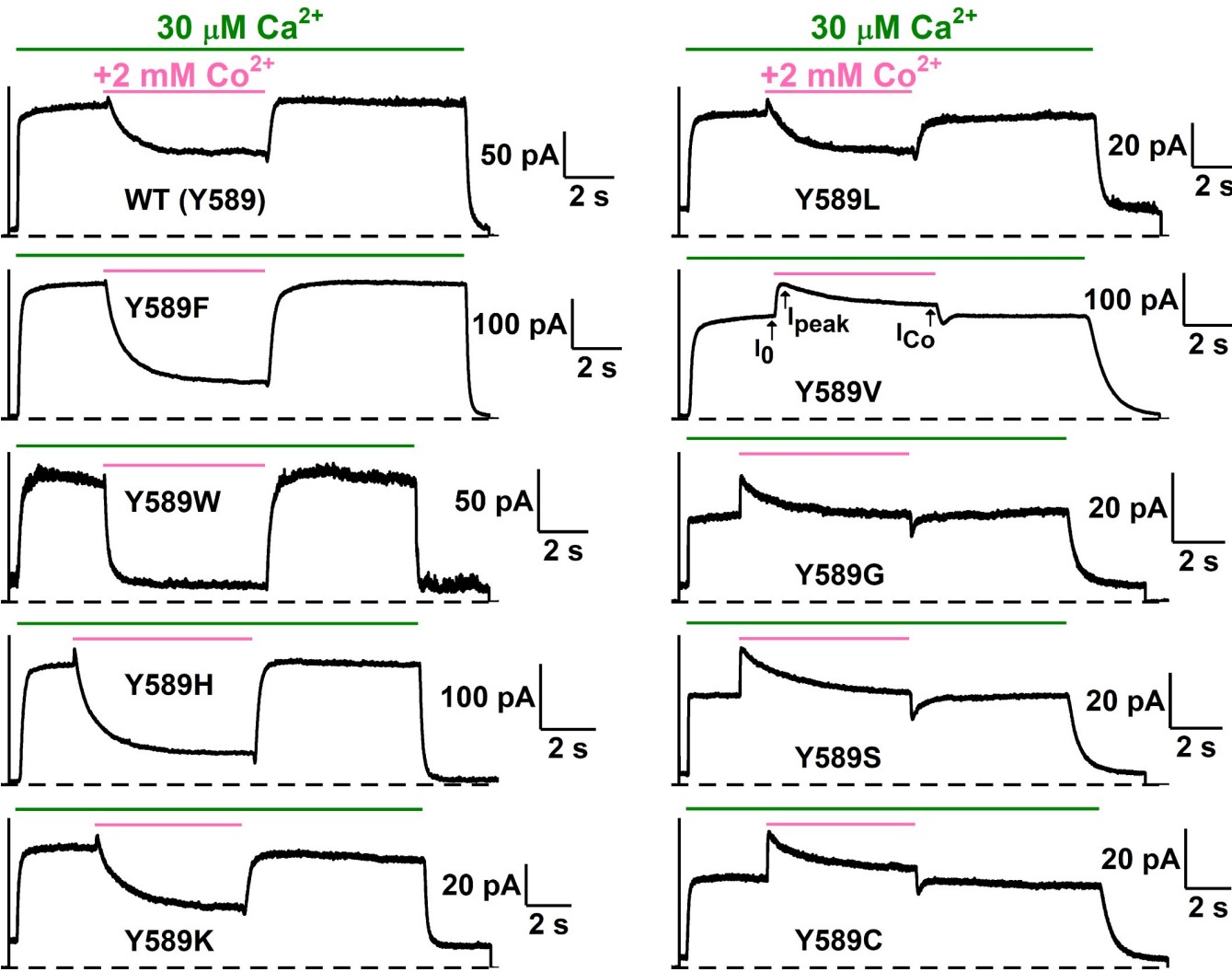

**Fig 6. Dependence of $Co^{2+}$ inhibition on the current reduction time ($\tau_{off}$) upon $Ca^{2+}$ washout.** Recording traces illustrate the effects of 2 mM $Co^{2+}$ on various Y589 mutants activated by 30 μM $[Ca^{2+}]_i$ at +20 mV. Dissociation rates of $Ca^{2+}$ were evaluated from the current reduction process upon $Ca^{2+}$ washout at the end of each recording by fitting the current reduction process with a single-exponential decay function (Eq 2). Notice that the degree of $Co^{2+}$ inhibition was reduced with the increase of the time constant ($\tau_{off}$) of the current reduction upon removing $[Ca^{2+}]_i$ (see correlation plot in the left panel of Fig 7A).

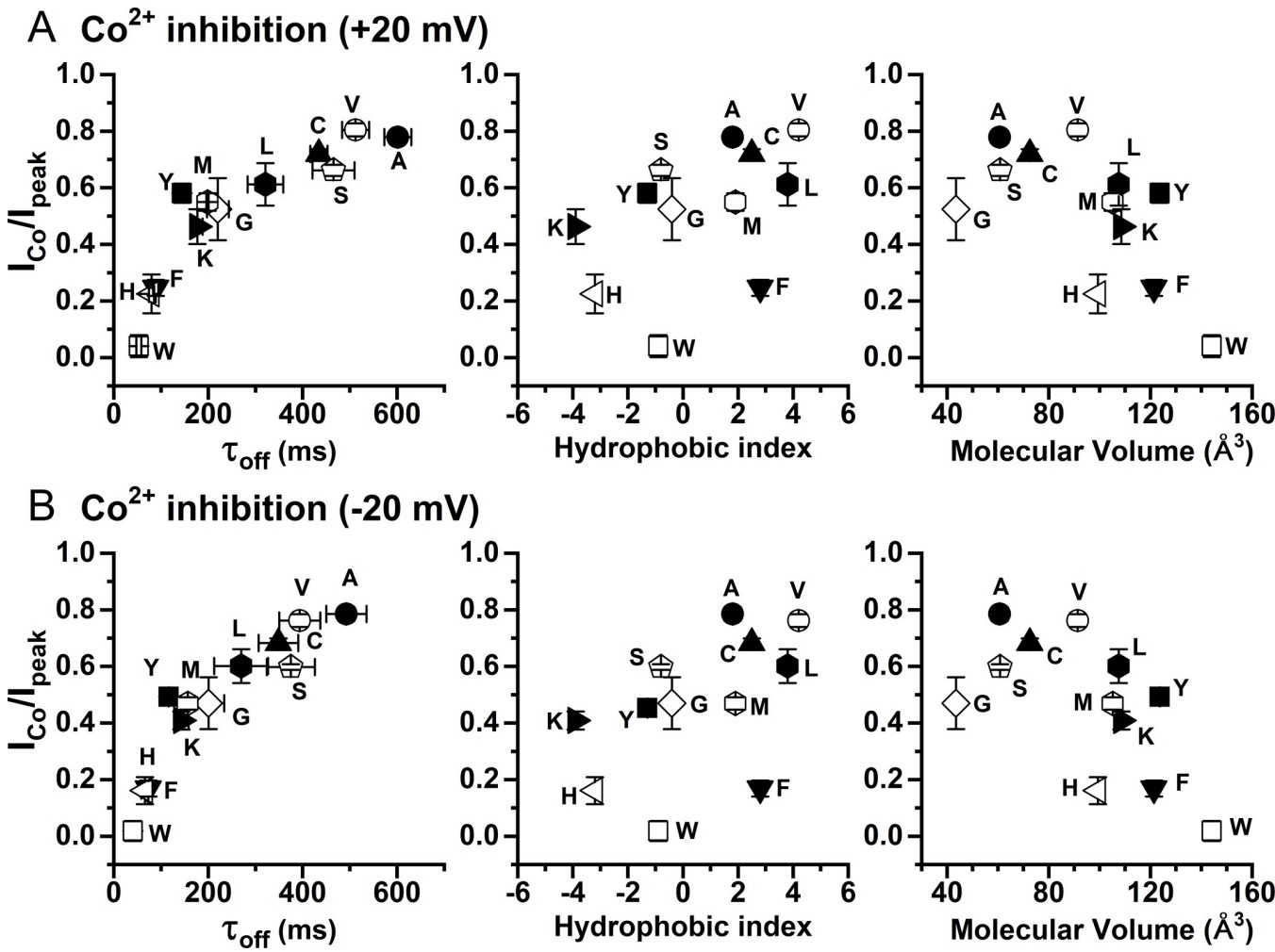

**Fig 7. Correlation of Co²⁺ inhibition with τ_off, the hydrophobicity index, or the molecular volume of the amino acid placed at position 589.** Here, experiments were similar to those shown in Fig 6. the remaining current fraction after 6-sec Co²⁺ application ($I_{Co}/I_{peak}$) was calculated and was plotted against the three different parameters. (A) Results obtained at +20 mV. (B) Results obtained at -20 mV. All data points were obtained from the effects of 2 mM $[Co^{2+}]_i$ on the WT TMEM16A current induced by 30 µM $[Ca^{2+}]_i$.

acid at position 589. Visual inspection of these correlation plots suggests that Co²⁺ inhibition decreases with the increase of τ_off (Fig 7A and 7B, left panel), while the correlations of Co²⁺ inhibition with the sidechain hydrophobicity (Fig 7A and 7B, middle panel) and with the molecular volume (Fig 7A and 7B, right panel) are weak. The inverse correlation of the potency of Co²⁺ inhibition with the Ca²⁺-dissociation rate supports the idea that Co²⁺ and Ca²⁺ compete for the high-affinity Ca²⁺-activation sites on the channel.

As shown in the original recording traces (for example, see Figs 2B or 6), intracellular Co²⁺ also potentiates the channel current. To extend data analyses, we also plot the degree of potentiation against the τ_off of current deactivation, the sidechain hydrophobicity, and the molecular volume of the introduced amino acid at position 589 (Fig 8). Unexpectedly, the potentiation was found to correlate best with the molecular volume of the amino acid—the Co²⁺ potentiation was larger in mutants with a smaller amino acid sidechain at position 589 (Fig 8A and 8B, right panel). On the other hand, the correlations between the degree of Co²⁺ potentiation with τ_off

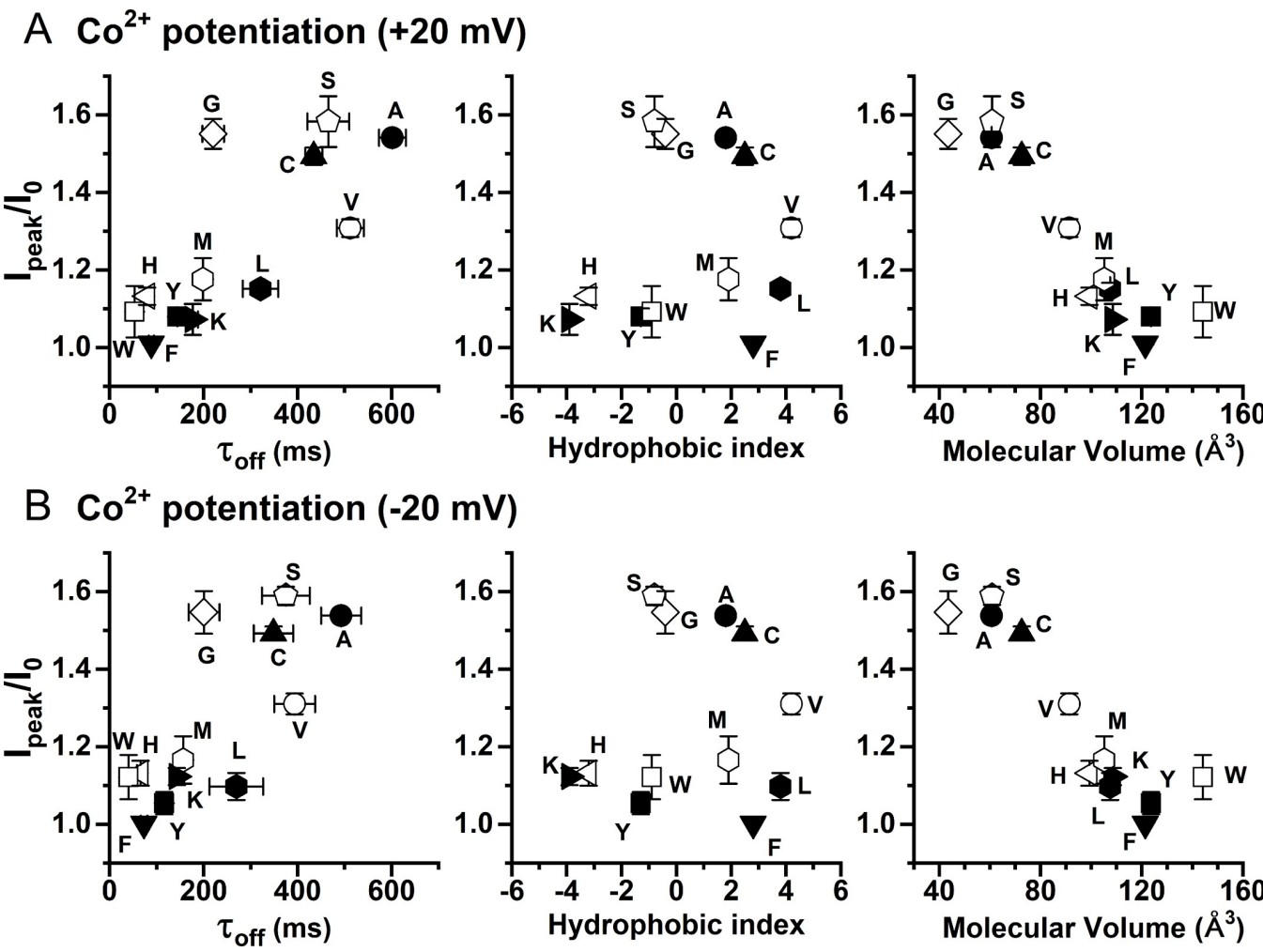

**Fig 8. Correlation of Co²⁺ potentiation with $\tau_{off}$, the hydrophobicity index, or the molecular volume of the amino acid at position 589.** (A) Results obtained at +20 mV. (B) Results obtained at -20 mV. All data points were obtained from the effects of 2 mM $[Co^{2+}]_i$ on the 30 μM $[Ca^{2+}]_i$-induced TMEM16A current. The degree of Co²⁺ potentiation was defined as $I_{peak}/I_0$. Notice a rough correlation between the potentiation and the volume of the amino acid placed at position 589—the larger the side-chain volume, the smaller the Co²⁺ potentiation.

(Fig 8A and 8B, left panel) and with the sidechain hydrophobicity (Fig 8A and 8B, middle panel) were weak.

To study the Co²⁺ potentiation more closely, we examined the concentration-dependent effect of Co²⁺. For the recording traces shown in Fig 9A, the WT TMEM16A currents were respectively induced by 100, 300 and 1000 μM $[Ca^{2+}]_i$. $[Co^{2+}]_i$ of various concentrations were applied at +20 mV (orange traces) and -20 mV (black traces), and the concentration-dependent Co²⁺ potentiation was shown in Fig 9B. The potentiation was minimal at sub-mM $[Co^{2+}]_i$. At the highest $[Co^{2+}]_i$ (20 mM), the potentiation was ~10% and ~35–45% of the control current at -20 mV and +20 mV, respectively. The results reveal a voltage dependence of Co²⁺ potentiation: the degree of potentiation at +20 mV is significantly larger than that at -20 mV, with this voltage-dependent difference most clearly observed at $[Co^{2+}]_i$ = 20 mM. It is also clear from these results that the affinity of Co²⁺ for the potentiation effect is low—the potentiation effect at +20 mV was not saturated even at 20 mM $[Co^{2+}]_i$! At -20 mV, the dose-dependent curve appears to have an apparent half-effective concentration of several mM.

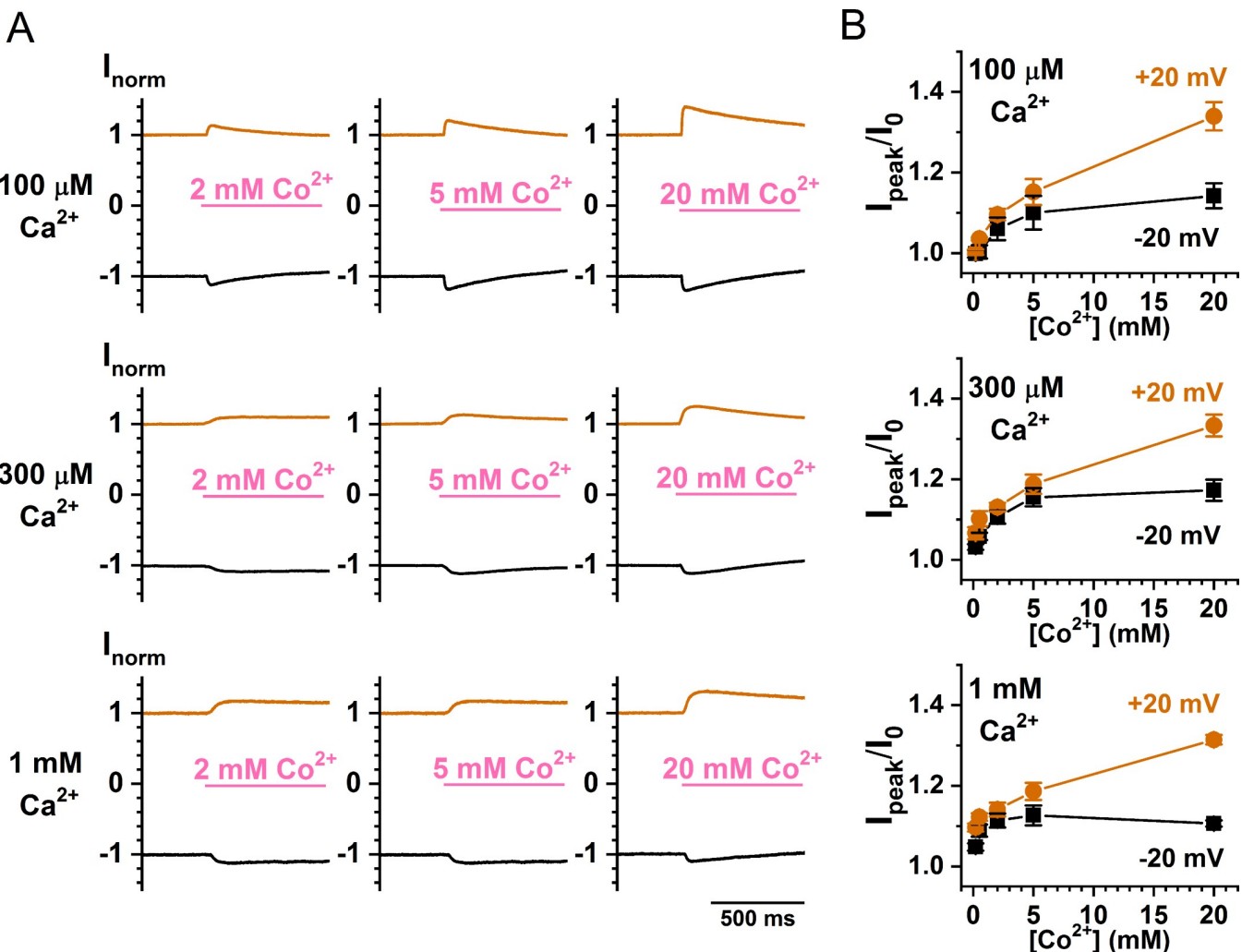

**Fig 9. Dose-dependent Co$^{2+}$ potentiation of the WT TMEM16A current.** (A) Recording traces depicting Co$^{2+}$ potentiation of the TMEM16A current activated by 100 μM, 300 μM, and 1 mM [Ca$^{2+}$]$_i$, respectively. To focus on the Co$^{2+}$ potentiation effect, only the traces from 450 ms before to 550 ms after the application of Co$^{2+}$ are shown. (B) Averaged Co$^{2+}$ potentiation of the WT TMEM16A current as a function of [Co$^{2+}$]$_i$. Notice the difference in the degree of potentiation between the experiments at -20 mV and +20 mV.

The observation that mM [Co$^{2+}$]$_i$ is required for potentiation is reminiscent of the finding that [Ca$^{2+}$]$_i$ in the mM concentration range induces more TMEM16A current even though the channel opening has already been saturated by low μM [Ca$^{2+}$]$_i$ [24, 29]. The dose-response curve of Ca$^{2+}$ activation of TMEM16A thus appeared as biphasic. Fig 10A shows an experiment on the WT TMEM16A, using the three-pulse protocol to compare the current induced by high [Ca$^{2+}$]$_i$ (2 or 20 mM) with that by 20 μM [Ca$^{2+}$]$_i$, a concentration thought to already saturate the high-affinity Ca$^{2+}$-activation sites [5, 30–32]. The recording traces reveal that the current induced by 2 or 20 mM [Ca$^{2+}$]$_i$ is significantly larger than the current induced by 20 μM [Ca$^{2+}$]$_i$. Furthermore, the current reduction process upon removing mM [Ca$^{2+}$]$_i$ shows two exponential decays, and the remaining current after the first exponential decay matches the amplitude of the current induced by 20 μM [Ca$^{2+}$]$_i$. These recording traces thus indicate that the current induced by mM [Ca$^{2+}$]$_i$ likely consists of two different components. We suspected that the extra current induced by mM [Ca$^{2+}$]$_i$ may have the same underlying

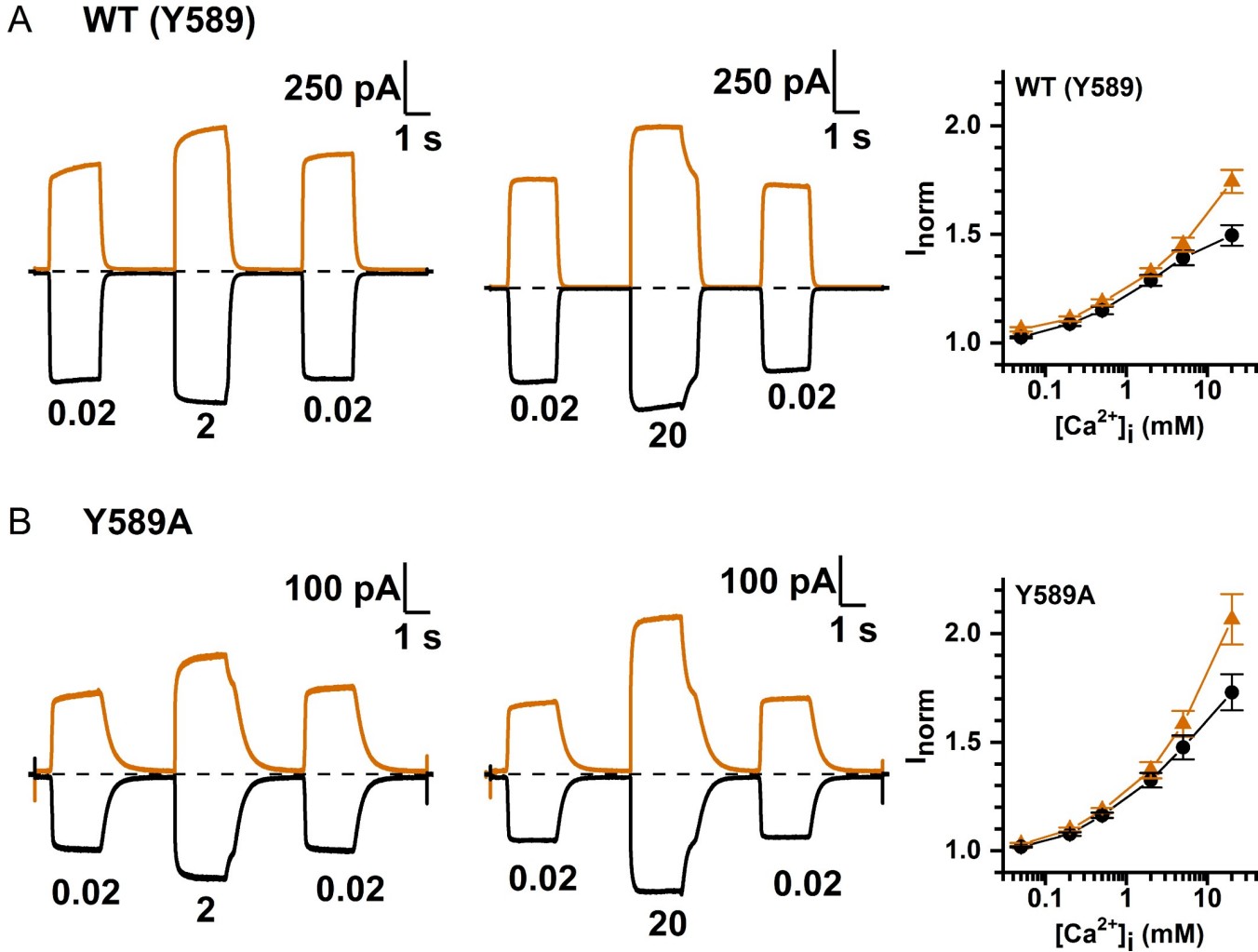

**Fig 10. Potentiation of TMEM16A current by mM concentrations of [Ca²⁺]ᵢ.** Recording traces of (A) WT TMEM16A and (B) Y589A. Both panels show a larger current induced by 2 mM (left panel) or 20 mM [Ca²⁺]ᵢ (middle panel) than that induced by 20 μM [Ca²⁺]ᵢ. Numbers below each pulse represent [Ca²⁺]ᵢ in mM. Orange and black traces are from +20 mV and -20 mV, respectively. Notice the fast and the slow current reduction upon washout of 2 or 20 mM [Ca²⁺]ᵢ. Right panels show dose-dependent Ca²⁺ potentiation of WT TMEM16A and Y589A at ±20 mV compared to the current induced by 20 μM [Ca²⁺]ᵢ. Current activated by various test [Ca²⁺]ᵢ (middle pulse) was normalized to the average of the control currents at 20 μM [Ca²⁺]ᵢ before and after the test [Ca²⁺]ᵢ.

mechanism for the Co²⁺ potentiation as both potentiation effects are mediated by relatively low-affinity binding of these two divalent cations (mM concentrations). Accordingly, we examined the high [Ca²⁺]ᵢ-induced current in various Y589 mutants more closely. The exemplary recording traces shown in Fig 10B indicate that the current reduction of the Y589A mutant upon washout of [Ca²⁺]ᵢ also consists of two components: a fast and a slow current-decaying component. The fraction of the fast current-decaying component (the low-affinity component) is larger than that in the WT channel, consistent with a larger Co²⁺ potentiation in Y589A than in the WT channel. The dose-response curves of Ca²⁺ potentiation in the WT channel and in various Y589 mutants obtained with high [Ca²⁺]ᵢ (from 50 μM to 20 mM) are depicted in Fig 11. In comparison with the potentiation on the WT channel (which has a tyrosine residue at 589 position), some mutants (such as Y589A, Y589C, and Y589W) show greater potentiation, and some have a similar degree of potentiation (such as Y589H and Y589L) while others show a smaller effect. To compare the potentiation by Ca²⁺ and Co²⁺, we plot the

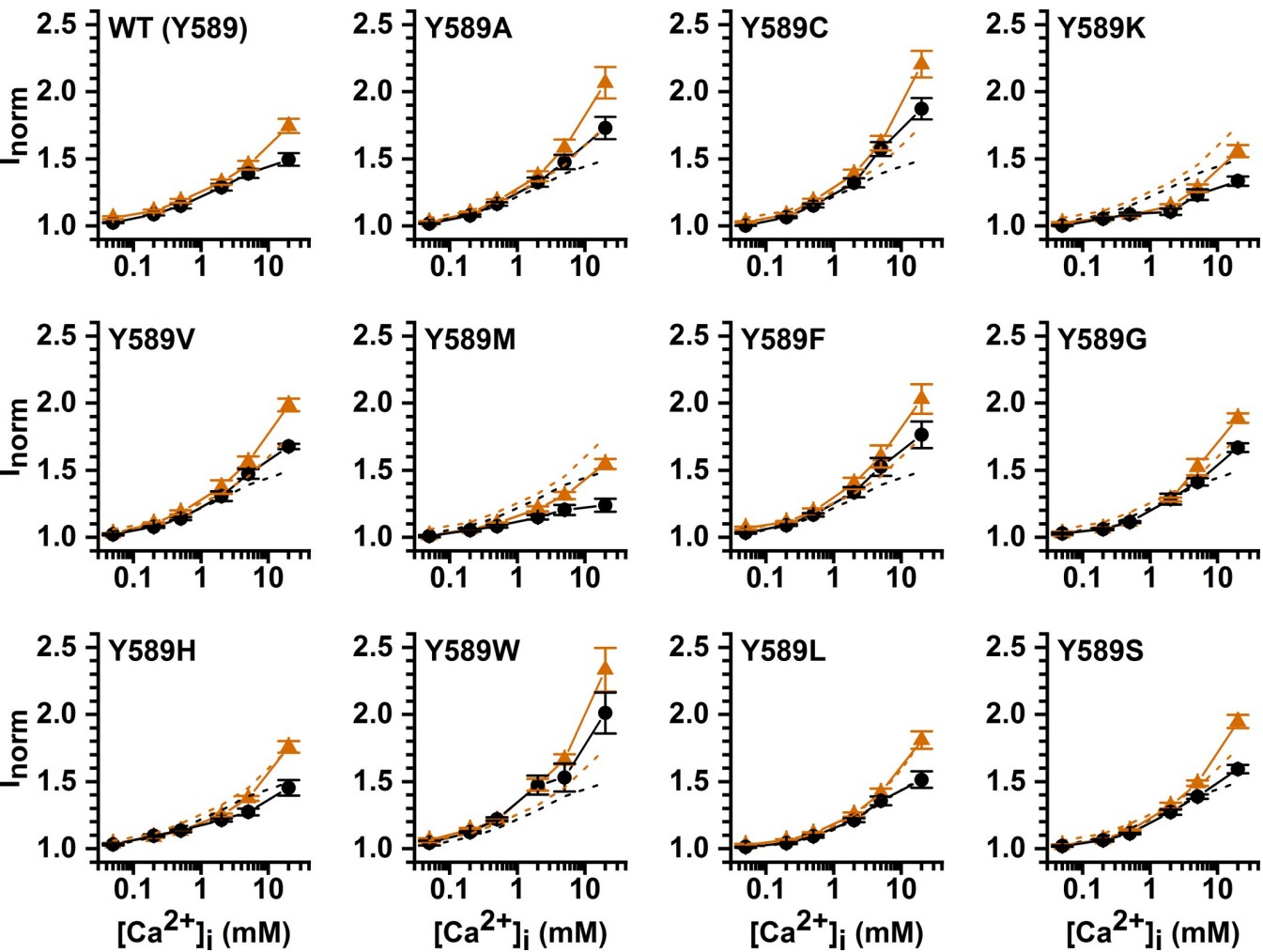

**Fig 11. Current potentiation in various Y589 mutants by high [Ca²⁺]ᵢ.** Dose-response curves at +20 mV (orange triangle) and -20 mV (black circle) were constructed from data obtained by the three-pulse protocol as that shown in Fig 10 (A & B). The dose-response curves of WT TMEM16A are plotted as dash curves in other panels for comparison.

potentiation by 20 mM [Ca²⁺]ᵢ (relative to the current induced by 20 μM [Ca²⁺]ᵢ) against the potentiation by 20 mM [Co²⁺]ᵢ (the current was activated by 300 μM [Ca²⁺]ᵢ) for all the Y589 mutants we have created (Fig 12). The results show that the Ca²⁺ potentiation and the Co²⁺ potentiation are roughly correlated with each other—the higher the Ca²⁺ potentiation, the larger the Co²⁺ potentiation. These results suggest that Ca²⁺ and Co²⁺ may act through the same mechanism to generate the potentiation effects.

## Discussion

TMEM16A is expressed in various tissues and plays many physiological roles, including mediating transepithelial anion transport [36, 37], modulating the mucin secretion and smooth muscle contraction in airways [38, 39], and controlling the motility of intestine [40, 41]. High-resolution structures of TMEM16 molecules show that multiple acidic residues use their side-chain carboxylates to coordinate the physiological ligand, Ca²⁺, in the Ca²⁺-binding sites [13–18], thus opening the channel. Although other alkaline earth divalent cations can bind to the

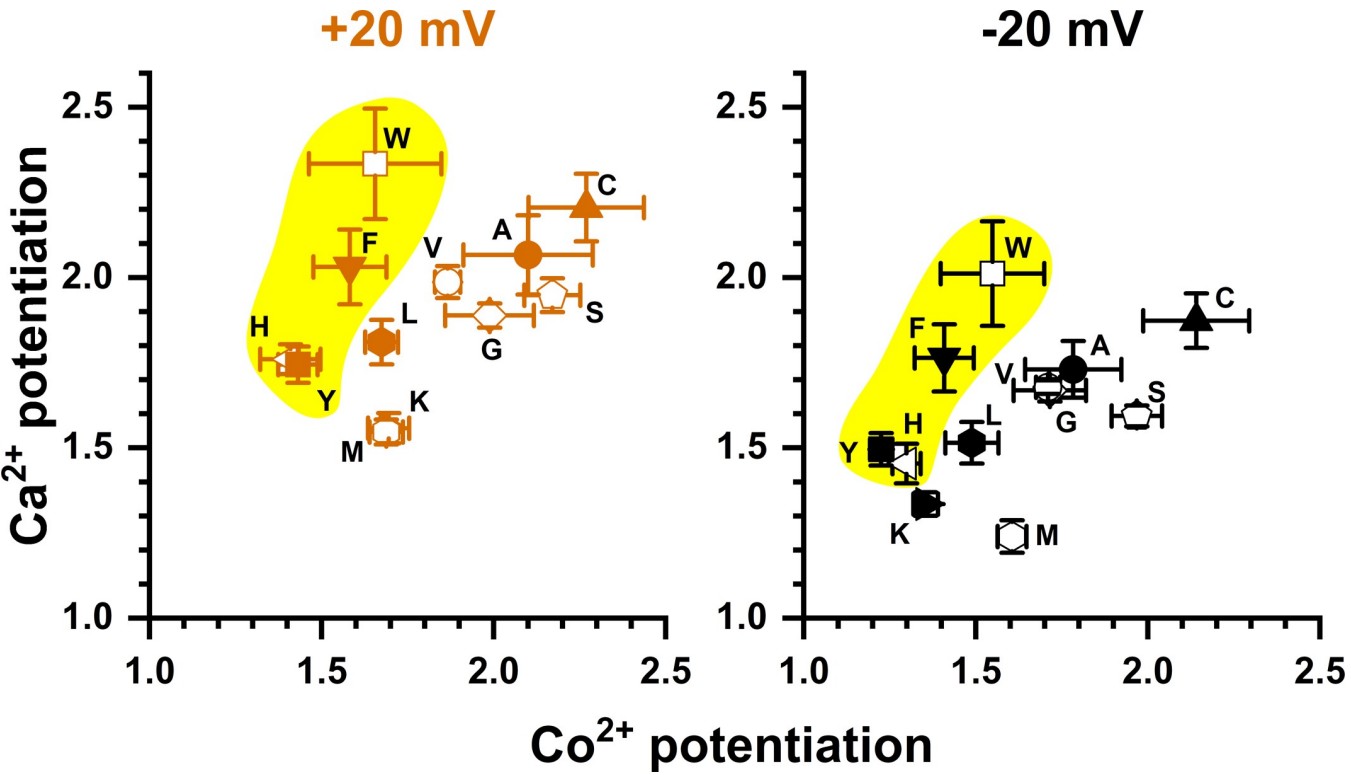

**Fig 12. Correlation of Ca²⁺ potentiation with Co²⁺ potentiation for various Y589 mutants.** Y-axis represents the potentiation by 20 mM $[Ca^{2+}]_i$ (namely, $I_{20\ mM}/I_{20\ \mu M}$) while X-axis is the potentiation of the $[Ca^{2+}]_i$ (300 μM)-induced current by 20 mM $[Co^{2+}]_i$. Aromatic mutants are highlighted in yellow.

activation sites, only $Sr^{2+}$ and $Ba^{2+}$ can induce current while $Mg^{2+}$ cannot. The binding of $Mg^{2+}$ to the activation sites was previously suggested through the observation that $Mg^{2+}$ shifts the $Ca^{2+}$-activation curve [30], suggesting a competition of $Mg^{2+}$ and $Ca^{2+}$ for the activation sites. In the absence of other divalent cations, the apparent affinities of $Ca^{2+}$, $Sr^{2+}$, and $Ba^{2+}$ in activating TMEM16A were shown to be in the range of ~0.5–1 μM, ~5–10 μM, and ~200–500 μM, respectively, while mM $[Mg^{2+}]$ is needed to shift the apparent $Ca^{2+}$ affinity by only twofold [30]. It is not surprising that the relative apparent affinities for these divalent cations in binding to TMEM16A appear to be qualitatively similar to those of their binding to EGTA, because the divalent cation-coordinating groups in TMEM16A and EGTA are carboxylates, providing a relatively high affinity for $Ca^{2+}$ binding. The activation of TMEM16A by sub-μM to low μM $[Ca^{2+}]_i$ is physiologically important because the channel can perfectly respond to the change of $[Ca^{2+}]_i$ from the resting state (~0.1 μM) to the excited state (sub μM to low μM) of cells.

In the present study, we demonstrate that another divalent cation, $Co^{2+}$, can also interact with TMEM16A, although intracellular $Co^{2+}$ up to 20 mM cannot induce current in WT TMEM16A (Fig 2A). Intracellular $Co^{2+}$, however, has two effects on the $Ca^{2+}$-induced TMEM16A current: an immediate potentiation of the $Ca^{2+}$-induced current followed by an inhibition of the current (Fig 2B). The degree of $Co^{2+}$ inhibition depends on the $[Ca^{2+}]_i$ used to induce the current—with a higher $[Ca^{2+}]_i$, a larger $[Co^{2+}]_i$ is required to exert the same degree of inhibition (Fig 3A). In fact, $Ca^{2+}$ shifts the dose-dependent $Co^{2+}$ inhibition curve in parallel towards the direction of higher $[Co^{2+}]_i$ (Fig 3B), and the rate of the $Ca^{2+}$ activation of TMEM16A decreases with the presence of $Co^{2+}$ (Fig 4A and 4B). Furthermore, mutant TMEM16A channels with a slower $Ca^{2+}$-dissociation rate from the activation sites show

weaker $Co^{2+}$ inhibition—the longer the current deactivation time ($\tau_{off}$) upon removing $[Ca^{2+}]_i$, the weaker the $Co^{2+}$ inhibition (Figs 6 and 7). We thus conclude that $Co^{2+}$ inhibition of the $Ca^{2+}$-induced current in TMEM16A likely results from the competition of $Co^{2+}$ with $Ca^{2+}$ for the high-affinity $Ca^{2+}$-activation sites. The apparent affinity of $Co^{2+}$ inhibition is quite high, likely due to its interaction with the sidechain carboxylate of multiple acidic residues in the $Ca^{2+}$-activation sites. For example, the $K_{1/2}$ of $Co^{2+}$ inhibition of the current induced by 2 μM $[Ca^{2+}]_i$ is only ~3–5 μM (Fig 3 and Table 1). In comparison, the $Mg^{2+}$ inhibition of the TMEM16A current activated by ~ 0.7 μM $[Ca^{2+}]_i$ has a $K_{1/2}$ of ~ 5 mM [30].

Besides inhibiting the $Ca^{2+}$-induced TMEM16A current, intracellular $Co^{2+}$ also potentiates the current, and this effect occurs before the inhibition appears. In all $Co^{2+}$ concentrations used, we cannot discern the difference between the rate of potentiation and the rate of solution exchange. Because the time courses of potentiation and inhibition can be clearly distinguished, and because the degree of potentiation does not significantly change in various $[Ca^{2+}]_i$ used in inducing current (Fig 9), $Co^{2+}$ potentiation is less likely to be a phenomenon mediated via high-affinity $Ca^{2+}$ activation sites. The current potentiation requires high $[Co^{2+}]_i$. Generating even a slight potentiation effect requires at least hundreds of μM of $[Co^{2+}]_i$. The potentiation appears to be voltage dependent: a clear difference in the degree of potentiation is observed between -20 mV and +20 mV when 20 mM $[Co^{2+}]_i$ was used to potentiate the current (Fig 9). Furthermore, the degree of $Co^{2+}$ potentiation is affected by mutation of Y589 (Figs 6 and 8), a pore residue. We thus suspect that the potentiation may result from an increase of the $Cl^-$ flux mediated by the binding of $Co^{2+}$ to the pore region.

Previous experiments have shown a biphasic $Ca^{2+}$ activation of TMEM16A [24, 29]. Namely, the TMEM16A activation is saturated at the concentration range from several μM to ~100–200 μM of $[Ca^{2+}]_i$. However, as $[Ca^{2+}]_i$ is further increased, more TMEM16A current can be induced [24, 29]. Recording traces in Fig 10A and 10B reveal that the current reduction upon removing mM $[Ca^{2+}]_i$ consists of a fast and a slow current-deactivation process, and the amplitude of the slower component is equivalent to the amplitude of the current induced by 20 μM $[Ca^{2+}]_i$. Thus, the fast-decaying component is the extra current induced by mM $[Ca^{2+}]_i$. Fig 10A and 10B also show that the amplitude of this low-affinity component in Y589A is larger than that in the WT channel, consistent with a larger $Co^{2+}$ potentiation in the Y589A mutant. We systematically compared over ten Y589 mutants for the degree of potentiation by $Ca^{2+}$ and $Co^{2+}$. The results indicate that the degree of potentiation by 20 mM $[Co^{2+}]_i$ (300 μM $[Ca^{2+}]_i$-activated current) roughly correlates with the degree of 20 mM $[Ca^{2+}]_i$-induced potentiation (relative to the 20 μM $[Ca^{2+}]_i$-activated current), except perhaps in two mutants, Y589W and Y589F (Fig 12). The less than perfect correlation in these two mutants could have two reasons. As shown in Fig 6, Y589W and Y589F have the shortest time constants ($\tau_{off}$) of current deactivation upon removing $[Ca^{2+}]_i$, so their affinity for $Ca^{2+}$ activation are low. Perhaps the 20 μM $[Ca^{2+}]_i$ used to induce the current in these two mutants was not a saturating concentration. Therefore, the current potentiation by 20 mM $[Ca^{2+}]_i$ (compared to the current induced by 20 μM $[Ca^{2+}]_i$) may include a further opening of the channel by a more saturating $[Ca^{2+}]_i$. A second possibility for the disproportionally higher $Ca^{2+}$ potentiation than the $Co^{2+}$ potentiation in Y589W and Y589F may be a true difference of the potentiation due to, for example, a different binding of these two divalent cations to mutants with an aromatic sidechain. The less than perfect correlation between $Ca^{2+}$ and $Co^{2+}$ potentiation in Y589F and Y589W does not undermine the observations that the affinities for the $Ca^{2+}$ and $Co^{2+}$ potentiation are low, and the degrees of the potentiation by these two cations are similar to each other.

It is intriguing that mutating the pore residue Y589 affects the degree of $Ca^{2+}$ and $Co^{2+}$ potentiation. The fact that the mutants Y589K, Y589L, and Y589M exhibit a similar degree of potentiation indicates that sidechain charge plays little role in the potentiation. The correlation

between sidechain hydrophobicity and the degree of potentiation is also weak. Rather, the potentiation appears to best correlate with the sidechain volume of the amino acid at position 589. We suspect that the sidechain of residue 589 probably does not directly interact with $Ca^{2+}$ or $Co^{2+}$. The potentiation requiring hundreds of μM or mM of $Ca^{2+}$ or $Co^{2+}$ further indicates that binding of these two divalent cations to generate the effect is of low affinity, and therefore could be non-specific. Recent experiments from our laboratory have revealed that introducing an aromatic residue at the Q559 position of TMEM16F, which corresponds to K584 of TMEM16A (a pore residue), significantly reduces the rundown of TMEM16F [28]. We speculated that this reduction of rundown in the Q559W mutant may involve phospholipids because membrane phosphatidylinositol diphosphate (PIP2) were shown to affect the rundown of TMEM16 molecules [42, 43]. Interaction of the fungus scramblase protein with phospholipids has been shown to thin the lipid bilayer near the transport pathway [16, 22]. Structures of TMEM16 molecules also suggest phospholipids may exist in the pore region [14–18], or may even form the wall of the substrate-transport pathways [44]. Interestingly, divalent cations are known to bind to phospholipids with binding affinities of mM or above [45–47]. It is thus possible that $Co^{2+}$ and $Ca^{2+}$ bind to the phospholipids located at the intracellular pore entrance to increase $Cl^-$ flux through the channel pore.

We and others have also shown that the sidechain charge from K584 (or K588 of the "a, c" alternatively spliced isoform) of TMEM16A electrostatically controls the $Cl^-$ flux [24, 28, 29]. The binding of divalent cations to the nearby regions would increase local $[Cl^-]$ via an electrostatic effect. The degree of the $Co^{2+}$ potentiation best correlates with the volume of the amino acid placed at position 589—a larger $Co^{2+}$ potentiation appears in the mutants with a smaller sidechain (Fig 8). Such a dependence on the sidechain volume of the introduced amino acid is reminiscent of previous studies on the *Torpedo* CLC-0 $Cl^-$ channel where the sidechain volume of a residue deep in the pore (E166 of CLC-0) affects the blocking affinity of amphiphlic pore blockers such as parachlorophenoxy acetate or octanoate [48, 49]. In those experiments, it was concluded that the charged end of the blockers "dock" at the pore entrance while the hydrophobic end of the blockers directly interact with the sidechain of the amino acid at the E166 position. If phospholipids contribute forming the pore wall of TMEM16A [44], the hydrophilic (or charged) end of phospholipids should be located at the pore entrance while the hydrophobic tail would be at a deeper position of the pore. Perhaps a smaller sidechain of residue 589 would allow the pore vestibule to accommodate more phospholipids for binding more divalent cations.

In summary, we have shown that intracellular $Co^{2+}$, like $Mg^{2+}$, competes with $Ca^{2+}$ for the channel activation sites and thus inhibits the $Ca^{2+}$-induced current in TMEM16A. $Co^{2+}$ at higher concentrations can also potentiate the $Ca^{2+}$-induced TMEM16A current. Potentiation of the TMEM16A current by $Co^{2+}$ is likely mediated by the same mechanism of the current potentiation by mM $[Ca^{2+}]_i$. We suggest that this potentiation may occur via the binding of divalent cations near or within the pore because of the voltage dependence of the potentiation and because pore residue mutations affect this potentiation. We suspect this potentiation effect may be related to the phospholipids near the intracellular pore region. It will require further experiments to refute or further support this conjecture that membrane phospholipids indeed involve in the $Ca^{2+}$ and $Co^{2+}$ potentiation of the TMEM16A current.

## Supporting information

**S1 Fig. Estimation of the concentration of contaminating $Ca^{2+}$.** (A) Recording traces of the currents of the TMEM16F Q559W mutant induced by various $[Ca^{2+}]_i$ using a three-pulse protocol. The dash line represents zero-current level. $V_m = +40$ mV. $[Ca^{2+}]_i$ in the first and the

third pulse was 2 mM, a saturating concentration. For the second pulse in the top trace, a solution containing neither EGTA nor any added $[Ca^{2+}]_i$ (called nominal zero-$Ca^{2+}$ solution) was used, while a solution containing a calculated free $[Ca^{2+}]_i$ of ~0.5 μM was used for the second pulse in the lower trace. The nominal zero-$Ca^{2+}$ solution used in the top trace contains only 140 mM NaCl and 10 mM HEPES (pH = 7.4 adjusted with NaOH), while the solution used for the second pulse in the lower trace contains extra 100 μM EGTA and an added $[Ca^{2+}]_i$ of 89.8 μM. Free $[Ca^{2+}]_i$ in the solution was calculated using the MaxChelator program (http://maxchelator.stanford.edu/CaEGTA-NIST.htm). (B) Normalized currents of a TMEM16F mutant, Q559W, activated by various $[Ca^{2+}]_i$. The amplitudes of all recorded currents were normalized to that induced by 2 mM $[Ca^{2+}]_i$. Solid black circles represent the data from using solutions containing calculated $[Ca^{2+}]_i$ of 0.5 μM, 2.0 μM, and 5.1 μM free $[Ca^{2+}]_i$ (n = 4–11), and the data match the dose-response curve for TMEM16F Q559W mutant reported in Nguyen et al. [28]. The solid pink circle represents the data from using the 0 $Ca^{2+}$ solution containing only the contaminating $[Ca^{2+}]_i$ (n = 11).
(TIF)

## Acknowledgments

We thank Dr. Robert Fairclough for helpful suggestions and critical reading of the manuscript.

## Author Contributions

**Conceptualization:** Dung M. Nguyen, Tsung-Yu Chen.

**Data curation:** Dung M. Nguyen, Louisa S. Chen, Grace Jeng, Tsung-Yu Chen.

**Formal analysis:** Dung M. Nguyen, Louisa S. Chen, Grace Jeng, Tsung-Yu Chen.

**Funding acquisition:** Tsung-Yu Chen.

**Investigation:** Dung M. Nguyen, Louisa S. Chen, Grace Jeng, Wei-Ping Yu, Tsung-Yu Chen.

**Methodology:** Dung M. Nguyen, Louisa S. Chen, Wei-Ping Yu, Tsung-Yu Chen.

**Project administration:** Tsung-Yu Chen.

**Resources:** Wei-Ping Yu, Tsung-Yu Chen.

**Software:** Tsung-Yu Chen.

**Supervision:** Tsung-Yu Chen.

**Validation:** Tsung-Yu Chen.

**Visualization:** Dung M. Nguyen, Tsung-Yu Chen.

**Writing – original draft:** Dung M. Nguyen, Tsung-Yu Chen.

**Writing – review & editing:** Dung M. Nguyen, Louisa S. Chen, Tsung-Yu Chen.

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
