## [Decision Letter · Decision Letter 0]

4 Mar 2020

PONE-D-20-03301

Cobalt Ion Interaction with TMEM16A Calcium-Activated Chloride Channel: Inhibition and Potentiation

PLOS ONE

Dear Dr. Tsung-Yu Chen,

Thank you for submitting your manuscript to PLOS ONE. After careful consideration, we feel that it has merit but does not fully meet PLOS ONE’s publication criteria as it currently stands. Therefore, we invite you to submit a revised version of the manuscript that addresses the points raised during the review process.

ACADEMIC EDITOR: Please correct your manuscript according to the criticism of the reviewers.

We would appreciate receiving your revised manuscript by Apr 18 2020 11:59PM. To enhance the reproducibility of your results, we recommend that if applicable you deposit your laboratory protocols in protocols.io, where a protocol can be assigned its own identifier (DOI) such that it can be cited independently in the future. For instructions see: http://journals.plos.org/plosone/s/submission-guidelines#loc-laboratory-protocols

We look forward to receiving your revised manuscript.

Kind regards,

Eugene A. Permyakov, Ph.D., Dr.Sci.

Academic Editor

PLOS ONE

Journal Requirements:

Reviewers' comments:

Reviewer's Responses to Questions

**Comments to the Author**

1. Is the manuscript technically sound, and do the data support the conclusions?

Reviewer #1: Partly

Reviewer #2: Yes

2. Has the statistical analysis been performed appropriately and rigorously? 

Reviewer #1: I Don't Know

Reviewer #2: Yes

3. Have the authors made all data underlying the findings in their manuscript fully available?

Reviewer #1: Yes

Reviewer #2: Yes

4. Is the manuscript presented in an intelligible fashion and written in standard English?

Reviewer #1: No

Reviewer #2: No

5. Review Comments to the Author

Reviewer #1: The submitted manuscript presents an in-dept investigation on the ion conductivity changes in a membrane Ca-activated chloride channel upon competitive binding of Co2+ ions. This protein is a transmembrane homodimer, each subunit having two high affinity binding sites for calcium. Correspondingly, the interpretation of just the current recordings upon the displacement of one type of metal by another type of metal is challenging. The authors have accumulated a respectable volume of current recordings in diverse conditions (including 10 mutant proteins). It would have definitely helped if the stability constants for cobalt ions were measured for all possible binding sites (and in the mutants). However, the complexity of lipid environment adds a whole extra layer of uncertainty. So, it is understandable that the data, as presented, do not make a completed picture.

Long story short, there are a number of questionable parts in the authors’ treatment of data, plus the English of the article is very heterogeneous. All that let me only to recommend a major revision of the manuscript. I have compiled the specific comments below.

Major comments

1. Ln. 101, 104: “…alter the affinity of Ca2+ in activating the channel…” — the affinity and channel activation should be discussed separately.

2. Ln. 169: “…the concentration of NaCl in the intracellular solution was reduced according to the extra [Cl-] from the added [CoCl2].” — such adjustments would have altered the ionic strength of the solution. Few words should be added on the possible impact on the observations.

3. “Data analysis” section is very poorly written, hard to read (compared e.g. to the Discussion, which is good).

4. Ln. 209: “was fit to a single-exponential function” — What kind of function is this? Apparently, it is valid only for the chosen concentration of EGTA?

5. The selection of a Langmuir-type equation for the data analysis is highly questionable. Why not employ standard competitive binding models?

6. Ln. 321-323, 340-341: “the correlation of Co2+ inhibition with the sidechain hydrophobicity is weak (Fig 7 A & B, middle panel). Inhibition appears to correlate with the molecular volume of the amino acid placed at position 589…” — I don’t see any correlations in these cases.

7. Fig. 8: The inverse correlation between the mutant residue volume and the current, could it be interpreted as a mechanical obstruction of the channel?

Minor and technical comments

1. Throughout (e.g. ln. 32, “…charged sidechain carboxylates to coordinate…”): ‘carboxylate’ is an anion of a carboxylic acid; the group is carboxyl (or ‘ionized carboxyl’ if one prefers).

2. Ln. 41: “…mutating a pore residue…” — in the abstract it’s preferable to be specific about which residue was mutated.

3. Ln. 64: “…conduction of ionic currents across lipid membranes. The physiological roles of the current conduction…” — it’s either ‘ion current’ or ‘ionic conduction’.

4. Ln. 68: “…are dimeric proteins consisting of two identical subunits…” — a proper term is ‘homodimeric’.

5. Ln. 96: “…each protein subunit consists of a set of Ca2+-binding sites…” — it doesn’t consist, but has exactly two binding sites, as follows from the published structures.

6. Ln. 106 : “…the binding alters the rectification of the Cl- flux through the channel pore by reducing the negative charge from these Ca2+-coordinating residues…” — not clear, should be rephrased.

7. Ln. 210: Fig 5C is cited not in sequence (the previous one was Fig 2B).

8. Ln. 514: What is CLC-0?

9. Ln. 521: Should be “vestibule”.

Reviewer #2: TMEM16A is a Ca2+-activated Cl- channel that plays key roles in diverse physiological processes including transepithelial Cl- transport. TMEM16A functional studies have been instrumental in deciphering the structure-function of the channel. For example, studies have shown that plotting Ca2+ activation of TMEM16A currents requires a Hill coefficient greater than 1, thereby consistent with multiple Ca2+ binding to activate TMEM16A currents. Moreover, functional studies demonstrated that the channel has two independent Cl- conducting pores.

Manuscript synopsis: Here, the authors explored TMEM16A gating using the divalent Co2+. Nguyen et al. report that Co2+ is a competitive inhibitor for the Ca2+ activation of TMEM16A. They also found that at high concentrations, Co2+ can potentiate Ca2+-evoked Cl- currents. These results were supported by inside-out patch experiments using intracellular Ca2+ application to activate TMEM16A, and application Co2+ to inhibit or potentiate these Ca2+-evoked Cl- currents. By varying concentrations of Ca2+ and Co2+ that were applied, the authors further characterize how Ca2+ and Co2+ regulate TMEM16A conducted Cl- currents. Additional experiments include mutations of the pore residue Y589 which were used to describe the relationship of Co2+ inhibition to amino hydrophobicity, volume, as well as the deactivation kinetics of the TMEM16A channel.

The experiments presented in this manuscript clearly demonstrate inhibition and potentiation Co2+ on TMEM16A Ca2+-evoked Cl- currents. Additionally, these experiments showing the effect of Co2+ on TMEM16A Ca2+-evoked Cl- current are well described and easy to understand. However, this is not as true for the Y589, whose interpretation in the manuscript somewhat confusing. The data included in this manuscript are interesting and should be published. However, improving the writing – specifically motivating the experiments presented in figures 6-10 and discussing the interpretation of these data - will create a more impactful manuscript.

Major points:

1) The motivation for the experiments depicted in figures 6-10 is not easily accessible to the reader. Even though understanding the characteristics of TMEM16A itself is important for the field, the specific angle for why we need to learn more about the biphasic dose response of TMEM16A is not stated. Moreover, reasons why the Y589 mutant was chosen for these experiments, or why this residue was targeted for multiple mutations, even why the amino acids used for these mutations were chosen should be clarified. Perhaps addressing how these data fit with current understanding of TMEM16A gating, or will direct future experimental directions will help the reader.

2) The manuscript alludes to the importance of amino acid carboxylates (line 97) in coordinating Ca2+ but does not address this concept in the context of the Co2+ findings and the Y589 mutants. The importance of these Y589 experiments in addressing the biphasic dose-response is also not clear. Perhaps further discussion on how these mutants address the main question could help.

3) Lines 71-84 and Figure 1 highlight TMEM16F and its structural properties. It is not clear how this information adds to or sets up the motivation for the question being addressed in this manuscript.

Minor points:

1) Lines 42-43 refer to Ca2+ as potentiating TMEM16A and not activating. Are the data suggesting that Ca2+ plays two roles?

2) Correlations described in Figures 7 and 8 are not apparent expect for �off. Are there patterns or features of the selected amino acids that should be described to help with the interpretation? The data are difficult to appreciate as described currently. Additionally, how these data relate to the overall question is quite unclear.

3) The manuscript is written as though it is a follow up to another study in a way that makes the current form of this one difficult to grasp and appreciate. The data are interesting and present new information about TMEM16A, and providing more rationale for the experiments conducted will enhance the value of the data.

6. PLOS authors have the option to publish the peer review history of their article (what does this mean?). If published, this will include your full peer review and any attached files.

Reviewer #1: No

Reviewer #2: No

---

## [Author Response · Author response to Decision Letter 0]

30 Mar 2020

See information in the "Response to Reviewers" file.

---

## [Editor Report · Decision Letter 1]

2 Apr 2020

Cobalt Ion Interaction with TMEM16A Calcium-Activated Chloride Channel: Inhibition and Potentiation

PONE-D-20-03301R1

Dear Dr. Tsung-Yu Chen,

We are pleased to inform you that your manuscript has been judged scientifically suitable for publication and will be formally accepted for publication once it complies with all outstanding technical requirements.

With kind regards,

Eugene A. Permyakov, Ph.D., Dr.Sci.

Academic Editor

PLOS ONE
---

## [Editor Report · Acceptance letter]

6 Apr 2020

PONE-D-20-03301R1 

Cobalt Ion Interaction with TMEM16A Calcium-Activated Chloride Channel: Inhibition and Potentiation 

Dear Dr. Chen:

I am pleased to inform you that your manuscript has been deemed suitable for publication in PLOS ONE. Congratulations! Your manuscript is now with our production department. 

With kind regards,

on behalf of

Prof. Eugene A. Permyakov 

Academic Editor

PLOS ONE